# Examining the impact of a community-based exercise intervention on cardiorespiratory fitness, cardiovascular health, strength, flexibility and physical activity among adults living with HIV: A three-phased intervention study

Kelly K. O'Brien[1,2,3]*, Aileen M. Davis[1,2,4◦], Soo Chan Carusone[5◦], Lisa Avery[4,6◦], Ada Tang[7‡], Patricia Solomon[7‡], Rachel Aubry[1◦], Mehdi Zobeiry[8], Ivan Ilic[8], Zoran Pandovski[8], Ahmed M. Bayoumi[2,9,10◦]

**1** Department of Physical Therapy, University of Toronto, Toronto, Ontario, Canada, **2** Institute of Health Policy, Management and Evaluation (IHPME), University of Toronto, Toronto, Ontario, Canada, **3** Rehabilitation Sciences Institute (RSI), University of Toronto, Toronto, Ontario, Canada, **4** University Health Network, Toronto, Ontario, Canada, **5** Casey House, Toronto, Ontario, Canada, **6** Avery Information, Oshawa, Ontario, Canada, **7** School of Rehabilitation Science, Institute of Applied Health Sciences, McMaster University, Hamilton, Ontario, Canada, **8** Central Toronto YMCA, Toronto, Ontario, Canada, **9** MAP Centre for Urban Health Solutions, Li Ka Shing Knowledge Institute, St. Michael's Hospital, Toronto, Ontario, Canada, **10** Department of Medicine, Faculty of Medicine, University of Toronto, Toronto, Ontario, Canada

◦ These authors contributed equally to this work.
‡ These authors also contributed equally to this work.
* kelly.obrien@utoronto.ca

## Abstract

### Purpose

Our aim was to examine the impact of a community-based exercise (CBE) intervention on cardiorespiratory fitness, cardiovascular health, strength, flexibility, and physical activity outcomes among adults living with HIV.

### Methods

We conducted a longitudinal intervention study with community-dwelling adults living with HIV in Toronto, Canada. We measured cardiopulmonary fitness ($\dot{V}O_2$peak (primary outcome), heart rate, blood pressure), strength (grip strength, vertical jump, back extension, push-ups, curl ups), flexibility (sit and reach test), and self-reported physical activity bimonthly across three phases. Phase 1 included baseline monitoring (8 months); Phase 2 included the CBE Intervention (6 months): participants were asked to exercise (aerobic, strength, balance and flexibility training) for 90 minutes, 3 times/week, with weekly supervised coaching at a community-based fitness centre; and Phase 3 included follow-up (8 months) where participants were expected to continue with thrice weekly exercise

**Data Availability Statement:** All relevant data are within the paper and its Supporting information files.

**Funding:** This study was funded by the Canadian Institutes of Health Research (CIHR) HIV/AIDS Community-Based Research (CBR) Program (Funding Reference Number #CBR-139685; 160 Elgin Street, Ottawa, Ontario, Canada, K1A 0W9). https://cihr-irsc.gc.ca/e/193.html KKO was supported by a Canada Research Chair (CRC) in Episodic Disability and Rehabilitation thanks to funding from the Canada Research Chairs Program. AT was supported by a Clinician-Scientist Award (Phase II) from the Ontario Heart & Stroke Foundation (P-19-TA-1192). AMB was supported by the Fondation Alma and Baxter Ricard Chair in Inner City Health at St. Michael's Hospital and the University of Toronto.

**Competing interests:** The authors have declared that no competing interests exist.

independently. We used segmented regression (adjusted for baseline age and sex) to assess the change in trend (slope) among phases. Our main estimates of effect were the estimated change in slope, relative to baseline values, over the 6 month CBE intervention.

## Results

Of the 108 participants who initiated Phase 1, 80 (74%) started and 67/80 (84%) completed the intervention and 52/67 (77%) completed the study. Most participants were males (87%), with median age of 51 years (interquartile range (IQR): 45, 59). Participants reported a median of 4 concurrent health conditions in addition to HIV (IQR: 2,7). Participants attended a median of 18/25 (72%) weekly supervised sessions. Change in $\dot{V}O_2$peak attributed to the six-month Phase 2 CBE intervention was 0.56 ml/kg/min (95% Confidence Interval (CI): -1.27, 2.39). Significant effects of the intervention were observed for systolic blood pressure (-5.18 mmHg; 95% CI: -9.66, -0.71), push-ups (2.30 additional push-ups; 95% CI: 0.69, 3.91), curl ups (2.89 additional curl ups; 95% CI: 0.61, 5.17), and sit and reach test (1.74 cm; 95% CI: 0.21, 3.28). More participants engaged in self-reported strength (p<0.001) and flexibility (p = 0.02) physical activity at the end of intervention. During Phase 3 follow-up, there was a significant reduction in trend of benefits observed during the intervention phase for systolic blood pressure (1.52 mmHg/month; 95% CI: 0.67, 2.37) and sit and reach test (-0.42 cm/month; 95% CI: -0.68, -0.16).

## Conclusion

Adults living with HIV who engaged in this six-month CBE intervention demonstrated inconclusive results in relation to $\dot{V}O_2$peak, and potential improvements in other outcomes of cardiovascular health, strength, flexibility and self-reported physical activity. Future research should consider features tailored to promote uptake and sustained engagement in independent exercise among adults living with HIV.

## ClinicalTrials.gov Identifier

NCT02794415. https://clinicaltrials.gov/ct2/show/record/NCT02794415.

## Introduction

People living with HIV with access to combination antiretroviral therapy are living longer and can experience an increased prevalence of multimorbidity associated with HIV, medications and aging [1]. Globally, the number of people living with HIV over the age of 50 years tripled between 2000 and 2020 [2]. Adults aging with HIV can experience cardiovascular disease, diabetes, bone and joint disorders, diabetes, liver disease, and non-HIV-related cancers that have implications on health and survival [3–7]. Multimorbidity can have important implications on the physical and mental health and well-being of adults aging with HIV [8].

Exercise can reduce the risk of multimorbidity and improve health outcomes for adults living with HIV [9–11]. Systematic review evidence demonstrated that engaging in aerobic, resistive or combined aerobic and resistive exercise at least three times per week can improve cardiorespiratory fitness, strength and quality of life outcomes among adults living with HIV [12–14]. However, only 51% of people living with HIV met recommended guidelines for

moderate to vigorous physical activity (MVPA) of at least 150 minutes per week and people living with HIV were less active than people with other chronic illnesses [15]. Barriers to engaging in exercise for people living with HIV are multifactorial. Environmental (location, physical accessibility, cost), personal (multimorbidity, physical health, lack of knowledge or self-efficacy, anxiety), and social factors (competing priorities, caregiver responsibilities, and fear of social stigma) can prevent engagement in exercise for adults living with HIV [16–22]. Most systematic reviews did not consider these factors, and focused primarily on highly supervised therapeutic interventions by physiotherapists and exercise physiologists that may be unsustainable in community settings [12–14, 23].

Community-based exercise (CBE) involves individuals exercising under the supervision of a health or fitness instructor with the goal of promoting regular exercise in the community [24–26]. CBE can foster social interaction, support and encouragement to exercise while improving health outcomes (pain, function) among older adults, and persons with mobility issues, stroke, multiple sclerosis, and arthritis [24–30]. CBE may be an ideal self-management strategy to help people living with HIV manage health challenges associated with chronic conditions. However, the impact of CBE when translated into the community setting with adults living with HIV, and its sustainability over the long term are unknown.

Our aim was to examine the impact of a CBE intervention on cardiorespiratory fitness, cardiovascular health, strength, flexibility, and physical activity outcomes among adults living with HIV.

## Materials and methods

We conducted a single group prospective three-phased intervention study to measure the effect of a CBE intervention at the YMCA among adults with HIV in Toronto, Ontario [31, 32] from August 2016 to December 2018. Our primary outcome was cardiorespiratory fitness (peak oxygen consumption measured by $\dot{V}O_2$peak). Maximum or peak oxygen consumption ($\dot{V}O_2$peak) is a direct measure of cardiorespiratory fitness [33]. We considered 2 ml/kg/min to be a clinically important change in $\dot{V}O_2$peak, based on combination of clinical experience and interpretations of systematic review evidence examining exercise in adults living with HIV [13].

Secondary outcomes included resting heart rate (beats/min), blood pressure (systolic and diastolic; mmHg), strength (upper and lower extremity muscle strength and endurance), flexibility (hamstrings) and self-reported physical activity. These secondary outcomes were chosen because they are related to the primary outcome of interest, $\dot{V}O_2$peak, and commonly reported in previous systematic reviews on HIV and exercise [12, 13]. Outcomes were assessed every 8 weeks across three phases: 1) Phase 1 Baseline Monitoring (8 months); 2) Phase 2 CBE Intervention (6 months) and 3) Phase 3 Follow-Up (8 months), for a total of 12 time points across the 22-month study. The 8 month baseline and 8 month follow-up ensured equal timeframes prior to and after the intervention, ensured an equal number of data points in each phase (Baseline: T1–T4; Intervention: T5–T8; Follow-Up: T9-12).

The protocol was approved by the HIV/AIDS Research Ethics Board at the University of Toronto (Protocol #32910; ClinicalTrials.gov Identifier: NCT02794415). Written informed consent was received from all participants in the study. This study was part of a larger implementation science study using the RE-AIM (Reach-Evaluation-Adoption-Implementation-Maintenance) framework to investigate the process and outcomes of implementing CBE with adults living with HIV in the community [31, 34–37]. Our focus in this work is to examine the "E", effectiveness of the intervention, as a component within the RE-AIM framework [37].

## Participants

We recruited community-dwelling adults living with HIV (18 years and older) in Toronto, Canada who considered themselves medically stable and safe to participate in exercise. We recruited through a combination of HIV community-based organizations and the Toronto YMCA using recruitment posters as well as word of mouth. Interested individuals met with a research coordinator to review eligibility, which included completing the self-administered Physical Activity Readiness Questionnaire (PAR-Q+) [38], and reviewing the information letter and consent form. We asked participants who scored 'yes' to one or more items on the PAR-Q+ to contact their health provider to review and confirm their eligibility to participate in the study. Participant recruitment and ongoing enrollment occurred from July 2016 to January 2017. Details on the study design and protocol have been previously published [31].

## Intervention

Phase 1 (Baseline), served as the control phase for establishing baseline trends (change over time) of outcomes. Given the nature of the exercise intervention, participants were not blinded to the intervention.

In Phase 2 (CBE Intervention), participants were provided with a YMCA membership and met weekly with a certified personal training fitness coach at the YMCA (e.g. Canada YMCA Personal Trainer Certification, CAN FIT PRO Personal Training Specialist) for 6 months. We chose a 6-month CBE duration because it aligns with transtheoretical model evidence on the stages of behaviour change whereby the 'action' of practicing a new behaviour (exercise) lasts up to 24 weeks followed by the 'maintenance' stage during which to solidify a commitment to sustaining the new behavior (self-monitored exercise) [39].

Based on an individual participant's goals, interests and abilities, YMCA coaches established a personalized intervention that included a combination of aerobic, resistance, balance and flexibility training [40]. Participants were asked to exercise for 90 minutes, 3 times/week for 6 months. *Aerobic exercise* was prescribed 3 days/week at 60–70% peak heart rate for at least 30 minutes with type of activity based on participant choice. *Resistance exercise* involved ~8–10 exercises for major muscle groups, 3 days/week using resistance (weight) equal to 60–70% 1 repetition maximum with 10–12 repetitions each. *Balance training* included activities focused on motor skills (e.g. balance, agility, coordination and gait) for approximately 20–30 minutes, 3 days/week. *Flexibility exercise* included stretching major muscle groups in a static stretch for 10–30 seconds each with 2 repetitions, 3 days/week [41]. Participants had the option to engage in individual exercise (e.g. weight circuit training, endurance treadmill walking / jogging, stationary bike, swimming) and group-based exercise in the form of a class at the YMCA (e.g. circuit training, spinning, yoga, aerobics, dancing) [42]. Participants received in-person weekly supervised coaching at the YMCA to monitor progress and adjust exercise intensity (25 weeks). In addition, participants were invited to attend monthly group-based in-person education sessions related to self-management, HIV and health. Sessions topics included: HIV and exercise, complementary and alternative therapy, healthy eating tips, sleep health, mindfulness and stress reduction, neurocognitive health, HIV and rehabilitation.

In Phase 3 (Follow-up), participants were provided with an extension of their YMCA membership for 8 months and were encouraged to continue with thrice weekly exercise independently with no coaching supervision.

## Data collection

We examined the primary outcome of interest ($\dot{V}O_2$peak) and secondary outcomes related to the primary outcome of interest as measured by objective and self-reported physical health

assessments. Participants completed a combination of fitness and questionnaire assessments every eight weeks for 8 months to obtain five baseline assessments (T1–T5 across Phase 1), followed by three assessments during the six-month intervention phase (T6–T8 Phase 2), and four assessments in the 8-month follow-up phase (T9–T12 Phase 3), for a total of 12 time points across the 22-month data collection timeframe. Data collection occurred from August 2016 to December 2018. Given the nature of the exercise intervention, assessors were not blinded to the intervention.

## Fitness assessments

Fitness assessments were conducted at the Central Toronto YMCA Performance Centre by certified fitness personnel in accordance with the Canadian Society of Exercise Physiology (CSEP) fitness assessment guidelines [43].

**Cardiorespiratory fitness.** *Peak oxygen consumption ($\dot{V}O_2peak$; ml/kg/min)* was measured at the YMCA on a stationary bike (Monark 817, Monark, Sweden) with metabolic cart for measurement of breath-to-breath gas exchange (CardioCoach 9000, KORR Medical Technologies Inc., USA) using an incremental test protocol. During the exercise test, participants started at a work rate intensity of 50–100 Watts (W) and increased their intensity by 25 Watts per minute while maintaining a target cadence of 50 revolutions per minute. YMCA staff followed the American College of Sports Medicine guidelines assessment procedure and criteria for test termination [41, 44]. Some participants may have had difficulty reaching $\dot{V}O_2max$ during the assessment due to difficulty tolerating the face mask or episodes of illness affecting their ability to reach maximum effort. Hence, we referred to '$\dot{V}O_2peak$' rather than '$\dot{V}O_2max$' in our study.

**Cardiovascular health.** Resting Heart Rate was measured as beats per minute prior to the $\dot{V}O_2peak$ assessment. Resting Blood Pressure (mmHg) was measured in the sitting position, taken at the brachial artery following the American Heart Association recommendations [45], also prior to the $\dot{V}O_2peak$ assessment.

**Strength.** We measured upper and lower body strength and endurance including grip strength using a digital dynamometer (kilograms of force), lower limb power using the vertical jump test (height in cm), and muscle endurance using the back extension test (maximum seconds able to hold), push-ups (maximum number of repetitions), and partial curl ups (maximum number of repetitions up to 25 maximum) [43].

**Flexibility.** We assessed flexibility of the lower back and hamstring muscles with the sit and reach test and recorded the score to the nearest centimeter of the distance reached by the hand [46]. Strength and flexibility were assessed after $\dot{V}O_2peak$ assessment.

## Questionnaire assessments

Self-reported questionnaires were completed in-person electronically at the University of Toronto.

**Physical activity.** We electronically administered the Rapid Assessment of Physical Activity (RAPA) questionnaire [47] every 8 weeks using Qualtrics software. This 9-item questionnaire is comprised of two domains that assess engagement in physical activity over the past week. RAPA 1 scores indicate level and intensity of engagement in aerobic physical activity (score range: 1–5 with higher scores indicating greater engagement). RAPA 2 scores are categorial and indicate regular engagement in exercises of strength only (score = 1), flexibility only (2), both (3) or neither (0).

**Exercise dose.** In Phase 2 and 3, participants were asked to complete a weekly exercise questionnaire from home, administered electronically using Qualtrics to document their level

of engagement (dose) of exercise. We asked participants to indicate the total number of minutes they engaged in moderate to vigorous physical activity (MVPA) within the past week and total number of minutes engaged in vigorous physical activity within the past week. Questions about dosage were guided by the Canadian Physical Activity Guidelines (CPAGs), which uses number of minutes engaged in moderate to vigorous aerobic physical activity in the past week as a unit of measurement [48, 49].

**Demographic and health characteristics.**   Participants completed a demographic and health questionnaire administered electronically at Phase 1 study initiation (T1) with items including but not limited to age, sex, length of time since HIV diagnosis, ethnoracial background, antiretroviral use, and number and type of concurrent health conditions.

We developed the weekly exercise, and demographic and health questionnaires as a team to include items that comprised personal, health and exercise-related characteristics. We piloted the weekly exercise questionnaire and demographic and health questionnaire prior to implementation with members of the team.

## Statistical analysis

**Primary analyses.**   We compared measures of cardiorespiratory fitness, strength, flexibility, and physical activity during Phase 1 baseline (0–8 months; 5 time points (T) T1–T5), Phase 2 intervention (8–14 months; 3 time points; T6–T8) and Phase 3 post-intervention (14–22 months; 4 time points T9–T12) using linear mixed effects models.

**Primary outcome ($\dot{V}O_2peak$).**   We conducted a preliminary analysis, using only baseline data, to determine how age and sex affect $\dot{V}O_2peak$. Previous research has found that $\dot{V}O_2max$ is 27% lower in females compared with males and declines about 10% per decade [50]. We found that $\dot{V}O_2peak$ varied by age and sex, but there was no significant age by sex interaction, so this was not included in subsequent models.

To estimate the effect of the intervention on $\dot{V}O_2peak$ we fit a segmented mixed effect linear regression model. Linear segments were fit for each phase of the study (baseline, intervention and follow-up), with random effects for the baseline intercept and slope. Age and sex were included as model covariates. An intention to treat analysis was used, whereby we analyzed all participants as receiving six months of intervention, as intended. The outcome of interest was the expected change in $\dot{V}O2peak$ over the six month intervention, in excess of what was observed during the baseline phase. This was calculated as $6 \times (\beta_{intervention} - \beta_{baseline})$, where $\beta_{intervention}$ and $\beta_{baseline}$ represent the expected monthly change in $\dot{V}O_2peak$ during the intervention and baseline periods, respectively and the difference represents the change attributable to the intervention. Following the recommendation of Peters et al. [51] missing data were not imputed. Model fit was assessed by examining residual plots and normal Q-Q plots. The complete model is detailed in S1 File. The R statistical programming language [52] was used for all analyses and modelling was performed with the nlme package [53].

**Secondary outcomes.**   We used similar approaches to modelling outcomes of cardiovascular health (resting heart rate and blood pressure), strength (grip strength, vertical jump, back extension, push-ups, curl ups), flexibility (sit and reach) and physical activity (RAPA 1 aerobic score). We interpreted $p < 0.05$ in the model to indicate significant trends (slopes) during baseline and intervention phases and to indicate whether there was a significant change in trend (slope) between the intervention (Phase 2) and follow-up (Phase 3). If baseline differences in sex were not significant at $p < 0.1$, and if removing sex improved model fit (assessed with the Bayesian Information Criterion) then we did not adjust for sex in the model. For the categorical RAPA 2 strength and flexibility outcome we used McNemar's $\chi2$ test to determine if the number of people participating in strength or flexibility physical activity was different

immediately prior to (T5) and after the intervention (start of follow-up phase) (T8) and after the follow-up at end of study (T12).

**Post hoc analyses.**   We conducted the following additional post hoc exploratory analyses.

**Per protocol analysis.**   We conducted a per protocol analysis to investigate the actual impact of intervention (including coaching sessions that were rescheduled post 25 weeks) on the outcomes of interest. For this analysis, we re-defined the baseline phase as any time prior to the first supervised training session and intervention phase as the time between the first and last supervised training session. Measurements after the last training session were not examined for this analysis (no Phase 3).

**Exercise dose.**   To investigate the effect of exercise dose on outcomes, we conducted a sub group analysis with participants who recorded engaging in any exercise in their weekly web-based exercise questionnaires. To address issues of missing data across the exercise questionnaires, we decided, based on the relative stability of the exercise minutes within participants, to use the median number of minutes of moderate to vigorous physical activity and number of minutes of vigorous physical activity as proxies for treatment dose. Because it is reasonable to expect that a participant's fitness will improve more with increasing dose, an interaction term (time in treatment phase x dose) was used to model the change in outcome over the treatment period, thereby allowing the rate of change of fitness outcomes to vary by dose. To determine the expected effect of the intervention on a participant meeting the Canadian Physical Activity Guidelines (CPAGs) [49], the estimated dose effect (per minute of exercise) was multiplied by the recommended amount in the guidelines: at least 150 moderate or vigorous minutes of physical activity in the past week (Model A); and at least 75 vigorous minutes of physical activity in the past week (Model B).

**Sample size.**   Given our longitudinal design with 12 time points, to detect a level and trend change in our primary outcome, $\dot{V}O_2peak$, assuming an effect size of 1, equal preintervention and postintervention time periods, statistical significance $p < 0.05$, and an autoregression error time series model with lag 1, and autocorrelation estimate 0.3, we would expect a power of 0.80 [54–56]. We aimed to recruit 120 participants, with the goal for 75 participants to complete the study, based on our observed retention rate of approximately 60% in a pilot study. More details on our sample size estimation can be found in our protocol [31].

## Results

Of the 120 individuals recruited to the study, 108 provided consent and initiated the baseline monitoring phase; of these, 80 (74%) started and 67/80 (84%) completed the intervention; and 52/67 (77%) completed the study (Fig 1).

Among the 108 participants at study initiation, most were males (89%), the median age was 51 years (males: 51 years, Interquartile Range (IQR) 45, 59; females: 48 years, IQR: 43, 61), 37% were from racialized communities, and the median number of concurrent health conditions in addition to HIV was 5 (IQR: 2,7). Participants self-reported engaging in a median of 196 minutes (IQR: 120, 306) of moderate or vigorous physical activity in the past week during Phase 2 (intervention) and Phase 3 (follow-up) (Table 1). The median duration between Phase 2 assessments (T5 to T8) was 189 days (27 weeks). Participants who started the intervention attended a median of 18/25 (72%) of the weekly supervised coaching sessions. Thirty of 80 participants who started the intervention, rescheduled at least one of their coaching sessions beyond the 25 week intervention phase due to episodes of illness, travel or work commitments. There were no serious adverse events documented from the intervention in the study. Baseline demographic characteristics of the sample in the study at enrollment, start of intervention, end of intervention, and end of study are detailed in S1 Table.

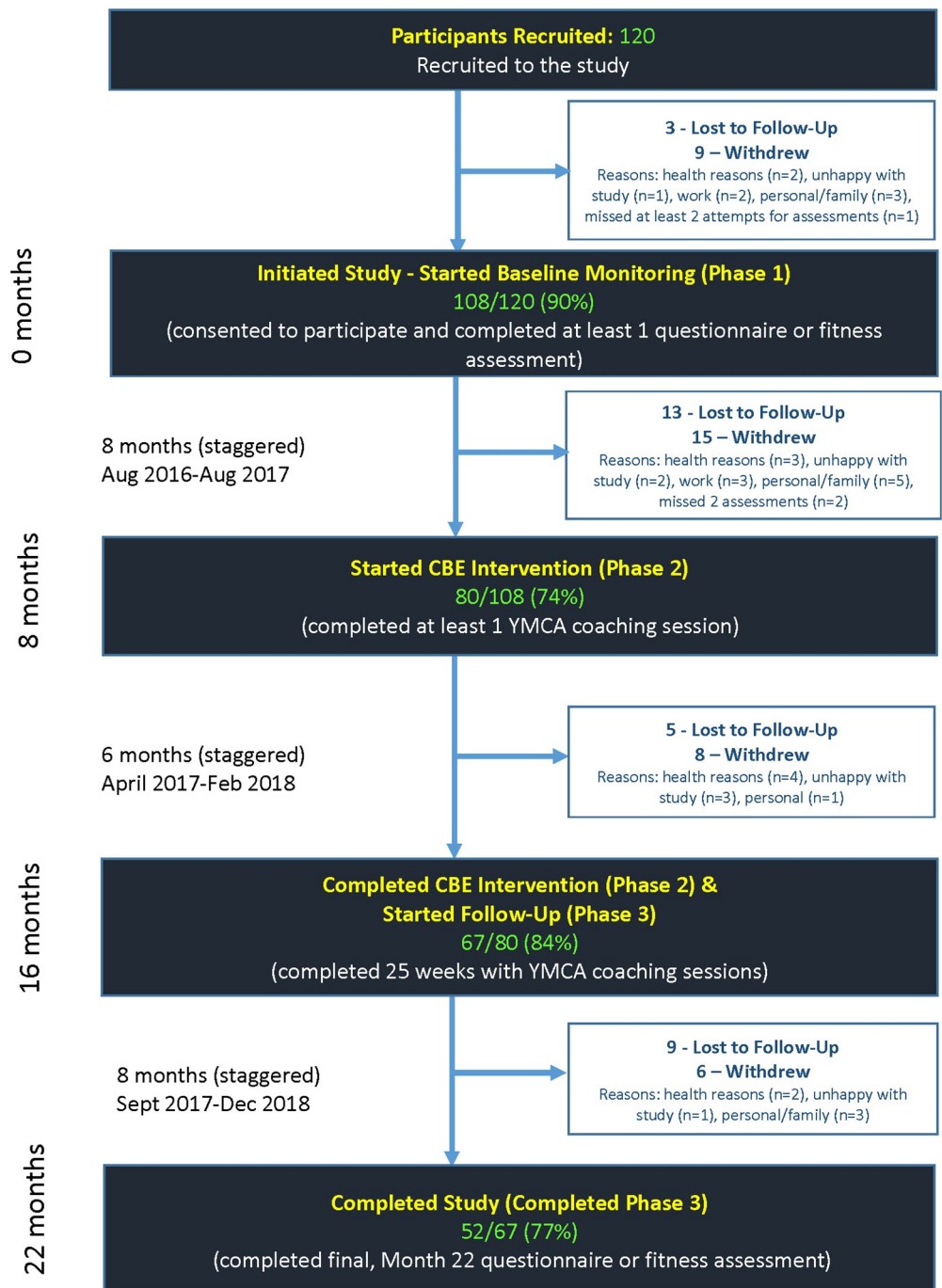

**Fig 1. Participant flow chart—Phase 1 (Baseline), Phase 2 (Intervention) and Phase 3 (Follow-up).**

## Primary outcome

$\dot{V}O_2$**peak.**   One hundred participants completed at least one $\dot{V}O_2$peak assessment and were included in the primary ($\dot{V}O_2$peak) analysis (90 males, 10 females) with a total of 689 observations across the three phases. Baseline mean (sd) $\dot{V}O_2$peak was 24.2 (8.0) ml/kg/min for males (n = 90) and 16.7 (4.1) ml/kg/min for females (n = 10). Results for the linear mixed

**Table 1. Characteristics of participants.**

| Characteristics at Study Initiation (Phase 1) | Number (%) |
|---|---|
| Median age (IQR) (n = 91) | 51 years (45, 59) |
| ≥ 50 years | 53 (49%) |
| **Sex** | |
| Male | 96 (89%) |
| Female | 12 (11%) |
| **Comorbidities** | |
| Median number of comorbidities[a] (IQR) | 5 (2,7) |
| Living with ≥2 comorbidities[a] | 89 (82%) |
| **Most commonly self-reported co-morbidities (>30%) included:** | |
| Mental health (e.g. depression, anxiety) | 52 (48%) |
| Joint pain (e.g. arthritis) | 44 (41%) |
| Muscle pain | 39 (36%) |
| Bone and joint disorder (osteopenia, osteoporosis, osteoarthritis) | 39 (36%) |
| **Median number of years since HIV diagnosis (25-75th percentile)** | 17 (8, 27) |
| **Current antiretroviral (HIV medication) use** | 107 (99%) |
| **Self-reported viral load undetectable** (<50 copies/mL) (n = 94) | 90 (83%) |
| **Self-reported current health status** | |
| Excellent | 8 (7%) |
| Very good or good | 80 (74%) |
| Fair | 18 (17%) |
| Poor | 2 (2%) |
| **Health status compared to previous year** | |
| Better now than 1 year ago | 50 (46%) |
| About the same as 1 year ago | 42 (39%) |
| Worse than 1 year ago | 16 (15%) |
| **Employed Full-Time or Part-Time** | 34 (31%) |
| **Smoking History** (n = 104) | |
| I currently smoke regularly or occasionally | 32 (30%) |
| I am a former smoker | 29 (30%) |
| I have never been a smoker | 40 (37%) |
| **Have Children** | 16 (15%) |
| **Live Alone** | 73 (68%) |
| **Gross average yearly income–CAD** (n = 107) | |
| ≤$20,000 | 57 (53%) |
| **Highest level of education** (n = 107) | |
| Completed high School/secondary school | 6 (6%) |
| Completed trade/technical school | 3 (3%) |
| Completed college | 24 (22%) |
| Completed university | 21 (20%) |
| Post-graduate education | 19 (18%) |
| **Exercise History (at start of exercise intervention)** (n = 83) | |
| "I currently exercise regularly and have done so for >6 months" | 30 (28%) |
| **Outcomes of Interest at Study Initiation[b]** | **Mean (SD)** |
| **$\dot{V}O_2$peak (ml/kg/min)** (n = 100) | |
| Males (n = 90) | 24.24 (±8.04) |
| Females (n = 10) | 16.70 (±4.14) |
| **Resting Heart Rate (beats per minute)** (n = 102) | 76.42 (±16.59) |

*(Continued)*

**Table 1.** (Continued)

| | |
|---|---|
| **Systolic Blood Pressure (mmHg)** | |
| Males and Females | 120.58 (±17.58) |
| **Diastolic Blood Pressure (mmHg)** | |
| Males (n = 91) | 76.55 (±11.83) |
| Females (n = 11) | 81.27 (±13.71) |
| **Grip Strength (kg)** | |
| Male (n = 91) | 78.47 (±17.74) |
| Female (n = 10) | 54.73 (±7.46) |
| **Vertical Jump Test (cm)** | |
| Male (n = 86) | 23.37 (±9.07) |
| Female (n = 10) | 15.55 (±7.31) |
| **Back Extension (sec)** | |
| Male (n = 87) | 67.06 (±43.05) |
| Female (n = 10) | 58.1 (28.06) |
| **Push Ups (number completed)** | |
| Male (n = 90) | 10.67 (±9.11) |
| Female (n = 10) | 8.50 (±6.47) |
| **Curl Ups (number completed up to 25 maximum)** | |
| Male (n = 88) | 15.98 (±10.27) |
| Female (n = 10) | 12.00 (±12.74) |
| **Flexibility—Sit and Reach Test (cm)** (n = 99) | 22.43 (±11.26) |
| **RAPA 1 –Aerobic physical activity (Range 1–5)** | |
| Male (n = 94) | 4.40 (±0.79) |
| Female (n = 11) | 3.91 (±0.70) |
| **RAPA 2 –Strength and Flexibility physical activity** (n = 108) | n (%) |
| 0 = None | 33 (31%) |
| 1 = Strength Only | 11 (10%) |
| 2 = Flexibility Only | 35 (32%) |
| 3 = Both Strength and Flexibility | 29 (27%) |
| **Self-Reported Physical Activity in Weekly Questionnaires**[c] | Median (IQR) |
| Minutes of Moderate or Vigorous Physical Activity in past week (Phase 2 and 3) | 196 (120, 306) |

Sample sizes: n = 108 (at enrollment) unless otherwise indicated; some sample sizes may not add up to total sample due to missingness on demographic and health questionnaire items; IQR: interquartile range;

[a]excluding HIV;

[b]reflects first measurement for each participant in the study; Outcomes of interest were reported separately for males and females if they differed and were adjusted for in the model.

[c]n = 45 participants who completed weekly web-based self-reported exercise questionnaires during Phase 2 (intervention) and Phase 3 (follow-up).

effect models of $\dot{V}O_2$peak over Phase 1 (baseline), Phase 2 (intervention) and Phase 3 (follow-up) are presented in Fig 2 and Table 2.

The estimated effect of the CBE intervention on $\dot{V}O_2$peak over the six-month intervention (Phase 2) taking the baseline (Phase 1) into account was 0.56 ml/kg/min (95% CI: -1.27, 2.39), with considerable variation of $\dot{V}O_2$peak within individuals over time (Table 2 and Fig 2). For the monthly rate of change (slope); during Phase 1, there was a small improvement in $\dot{V}O_2$peak (0.12ml/kg/min/month; 95% CI: -0.04, 0.27) followed by an increase in Phase 2

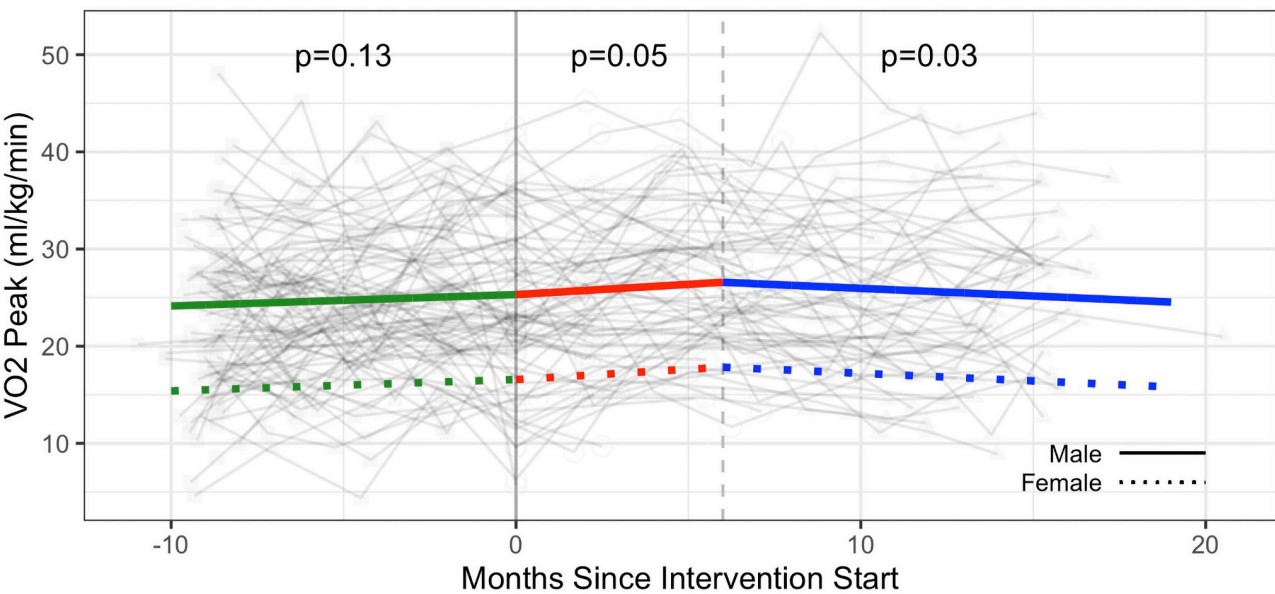

**Fig 2. Primary outcome—V̇O₂peak (ml/kg/min) trajectories over phase 1 (baseline), phase 2 (intervention) and phase 3 (follow-up monitoring); (n = 100 participants).** Vertical dotted line indicates end of phase 2 (intention to treat) at 6 months.

(0.21 ml/kg/min/month; 95% CI: 0.00, 0.42) and a small decline in Phase 3 (-0.16 ml/kg/min/month; 95% CI: -0.35, 0.03 [-0.16 ml/kg/min = 0.21 (Phase 2 slope) + (-0.37) (Phase 3 slope); Table 2]) (Table 2). The Phase 2 slope was not significantly different to the Phase 1 slope (p = 0.19) indicating the change in V̇O₂peak during the intervention phase was not

**Table 2. Results—Primary outcome—Cardiorespiratory fitness—V̇O₂peak.**

**V̇O₂peak (ml/kg/min)**

**Estimated Intervention Effect over 6 month Intervention Phase:**

0.56 ml/kg/min (95% CI: -1.27, 2.39)

Number of Observations: 689; Sample size: 100 (90 males; 10 females)

| Parameter | Fixed Effects | | | Random Effects (SD) |
|---|---|---|---|---|
| | **Estimate** | **95% Confidence Interval** | **p-value** | |
| Intercept (50yr old male) | 25.311 | 23.900, 26.721 | <0.001 | 5.67 |
| Age Effect | -0.138 | -0.248, -0.02 | 0.014 | - - |
| Sex Effect | -8.741 | -12.953, -4.529 | <0.001 | - - |
| Phase 1: Baseline slope (change per month) | 0.118 | -0.035, 0.270 | 0.130 | 0.40 |
| Phase 2: Intervention slope (change per month) | 0.211 | 0.004, 0.418 | 0.046[b] | 0.26 |
| Phase 3: Difference in Follow-up (Phase 3) and Intervention (Phase 2) slope[a] (change per month) | -0.368 | -0.704, -0.032 | 0.032 | <0.01 |
| Residual | - - | - - | - - | 4.15 |

Slopes are change in outcome over one month.

[a] Estimate refers to the difference in slope between the follow-up and intervention phase; Phase 3 (Follow-Up) slope is the sum of Phase 2 (intervention) slope and Phase 3 (difference in Follow-Up and Intervention slope): (0.211) + (-0.368) = -0.16 ml/kg/min.

[b]Significant trend in Phase 2 (intervention) slope versus 0; *Estimated intervention effect over intervention*: (0.211–0.118)*6 months = 0.56 ml/kg/min; *Interpretation*: Fixed effects indicate the average population level effect while random effects indicate the expected variability across people.

For instance, average V̇O₂peak value was 25.3 ml/kg/min at the beginning of the intervention with a 95% confidence interval of (23.9, 26.7), which refers to our level of confidence in the population average. The estimated random effect was 5.7 ml/kg/min, which indicates the variability in V̇O₂peak values in the sample population.

significantly different to the change during baseline (not shown). Individual trajectory plots of $\dot{V}O_2$peak for each participant are detailed in S1 Fig.

## Secondary outcomes

Table 3 summarizes the estimated effect of the CBE intervention (Phase 2) over and above any changes in baseline (Phase 1) for secondary outcomes. Results for secondary outcomes comparing Phase 1 (baseline), Phase 2 (intervention) and Phase 3 (follow-up) are detailed in S2 Table.

**Cardiovascular health.** The change in systolic blood pressure attributable to the intervention over 6 months was a decrease of 5.18 mmHg (95% CI: -9.66, -0.71); the monthly change during Phase 2 was -0.78 mmHg (p<0.01) and during Phase 3 was 0.74 mmHg (p<0.001; comparison with Phase 2) (Fig 3 and S2 Table).

Estimated intervention effects were not statistically significant for resting heart rate and diastolic blood pressure. There was no statistically significant monthly change in resting heart rate across all three phases, and a statistically significant monthly increase of 0.40 mmHg in diastolic blood pressure in Phase 3 (p = 0.02 comparison with Phase 2 (Figs 4 and 5 and S2 Table).

**Strength & flexibility.** The intervention was associated with a significant increase in the number of push ups (additional 2.30 push ups; 95% CI: 0.69, 3.91), the number of curl ups (+2.89 curl ups; 95% CI: 0.61, 5.17), and the sit and reach test distance (measure of flexibility) (1.74 cm; 95% CI: 0.21, 3.28) (Table 3, Figs 6–8 and S2 Table).

Estimated intervention effects were not statistically significant for grip strength and duration of back extension (Table 3, Figs 9–11 and S2 Table).

**Self-reported physical activity.** There were statistically significant increases in self-reported physical activity over the six-month intervention (Phase 2). For RAPA 1 (aerobic), participants demonstrated a statistically significant monthly increase in scores (0.04 points / month on scale 1–5) in the Phase 2 intervention (p<0.001), followed by a statistically

**Table 3. Secondary outcomes—Overall estimated effect of CBE intervention—Intention to treat analysis—Estimated effects of the six-month community-based exercise (CBE) intervention (Phase 2) after taking the eight-month baseline monitoring phase (Phase 1) into account.**

| Secondary Outcomes | Sample Size (n) | Number of Observations | Overall Effect Estimate (95% CI) |
|---|---|---|---|
| **Cardiorespiratory Fitness** | | | |
| Resting heart rate (bpm) | 102 | 718 | -0.75 (-3.97, 2.48) |
| Resting diastolic blood pressure (mmHg) | 102 | 724 | -0.68 (-3.78, 2.43) |
| Resting systolic blood pressure (mmHg) | 102 | 724 | -5.18 (-9.66, -0.71)[a] |
| **Strength and Flexibility** | | | |
| Upper extremity–Grip strength (kg) | 101 | 721 | -1.41 (-3.62, 0.81) |
| Lower extremity–Vertical jump test (cm) | 96 | 656 | -3.06 (-4.57, -1.54) |
| Back extension (seconds) | 97 | 667 | -6.90 (-16.32, 2.52) |
| Push ups (number completed) | 100 | 695 | 2.30 (0.69, 3.91)[a] |
| Curl ups (number completed) | 98 | 688 | 2.89 (0.61, 5.17)[a] |
| Flexibility–sit and reach test (cm) | 99 | 698 | 1.74 (0.21, 3.28)[a] |
| **Self-Reported Physical Activity** | | | |
| RAPA Aerobic scale | 105 | 805 | 0.16 (-0.03, 0.35) |

[a] Significant overall estimated treatment effect over the six-month intervention after taking the baseline monitoring into account;

$\dot{V}O_2$peak: peak oxygen consumption; RAPA: Rapid Assessment of Physical Activity; RAPA Aerobic (range: 1–5); CI: Confidence Interval.

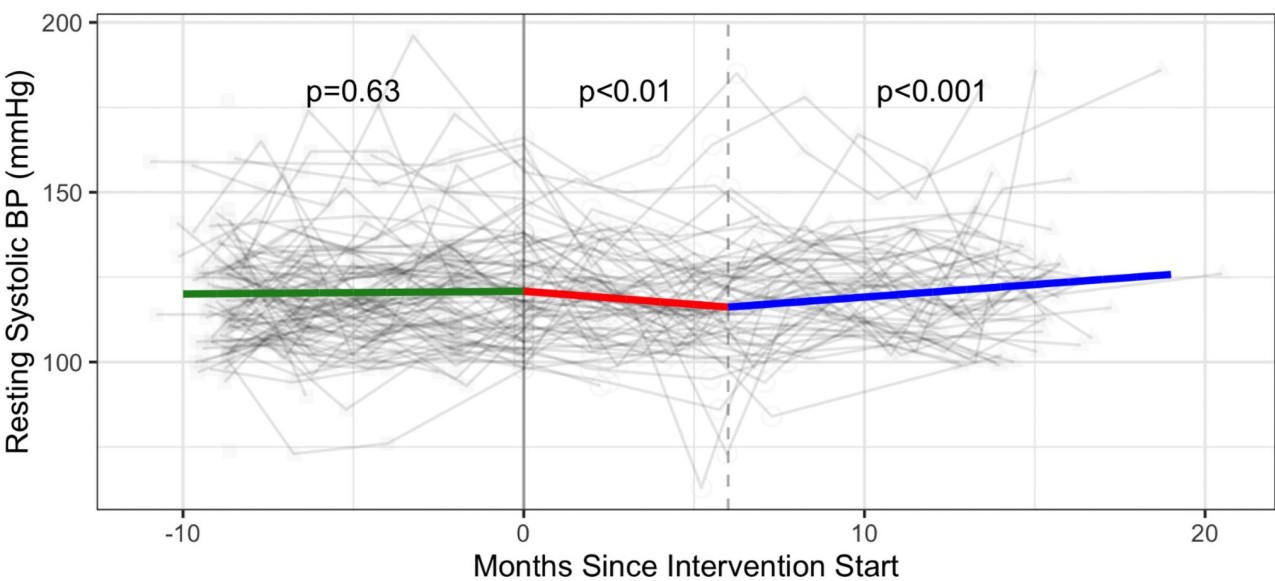

**Fig 3. Cardiovascular health—Systolic blood pressure (mmHg) trajectories over phase 1 (baseline), phase 2 (intervention) and phase 3 (follow-up monitoring); (n = 102 participants).** Vertical dotted line indicates end of phase 2 (intention to treat) at 6 months; BP: blood pressure.

significant decrease in scores in Phase 3 follow-up (-0.01 point / month) compared with Phase 2 (p<0.01) (Fig 12 and S2 Table).

For RAPA 2 (strength and flexibility) scores, 28 of the 33 (82%) participants not engaging in strength activity at the start of the intervention (T5) were engaged at the end of intervention (T8), while 3 of the 27 (11%) who engaged at T5 had stopped by T8 (McNemar's $\chi^2$ = 18.6, p<0.001). For flexibility 15/22 (68%) started flexibility training while 4/38 (11%) stopped (McNemar's $\chi^2$ = 5.3, p = 0.02) indicating that more participants engaged in strength and

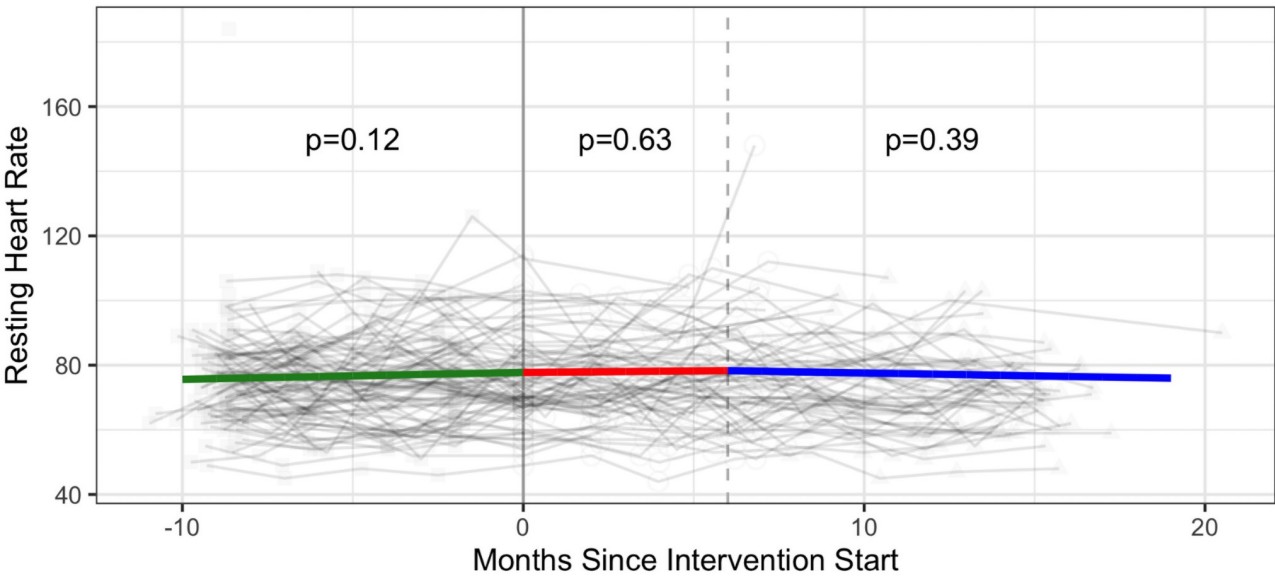

**Fig 4. Cardiovascular health—Resting heart rate (beats per minute) trajectories over phase 1 (baseline), phase 2 (intervention) and phase 3 (follow-up monitoring); (n = 102 participants).** Vertical dotted line indicates end of phase 2 (intention to treat) at 6 months.

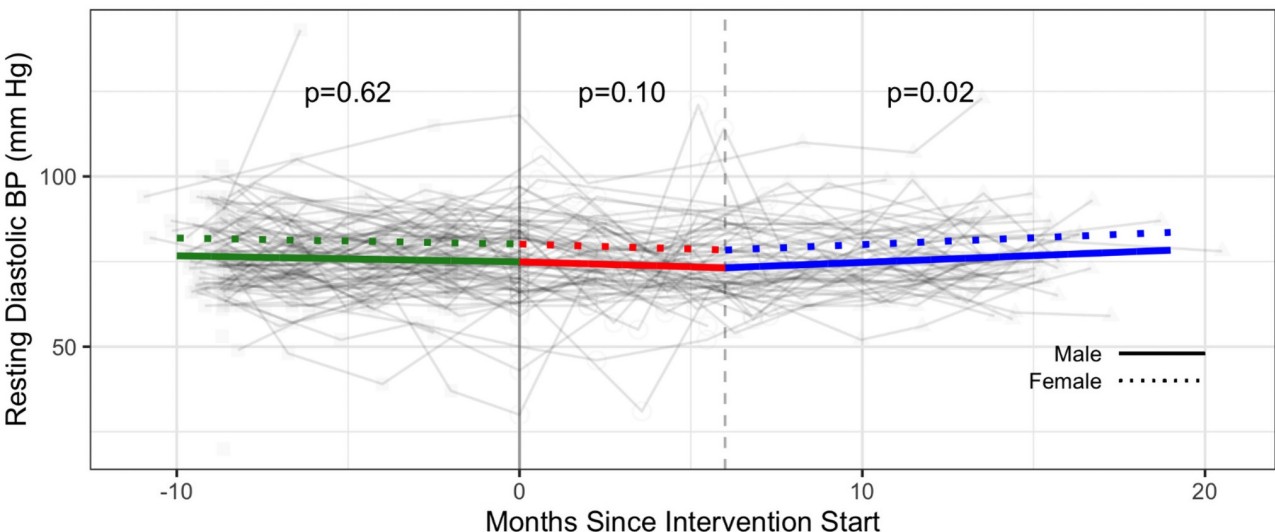

**Fig 5. Cardiovascular health—Diastolic blood pressure (mmHg) trajectories over phase 1 (baseline), phase 2 (intervention) and phase 3 (follow-up monitoring); (n = 102 participants).** Vertical dotted line indicates end of phase 2 (intention to treat) at 6 months; BP: blood pressure.

flexibility physical activity after the intervention versus prior to it. Fig 12 illustrates the proportion of participants engaged in strength and flexibility during each phase of the study (Fig 13 and S3 Table).

**Post hoc analysis.** Results for the exploratory post hoc (per protocol and exercise dose adjusted) analyses are detailed in S4–S6 Tables. S4 Table outlines the overall estimated effect of the six-month CBE intervention (Phase 2) over and above the baseline (Phase 1) for each outcome across all analyses (intention to treat and exploratory).

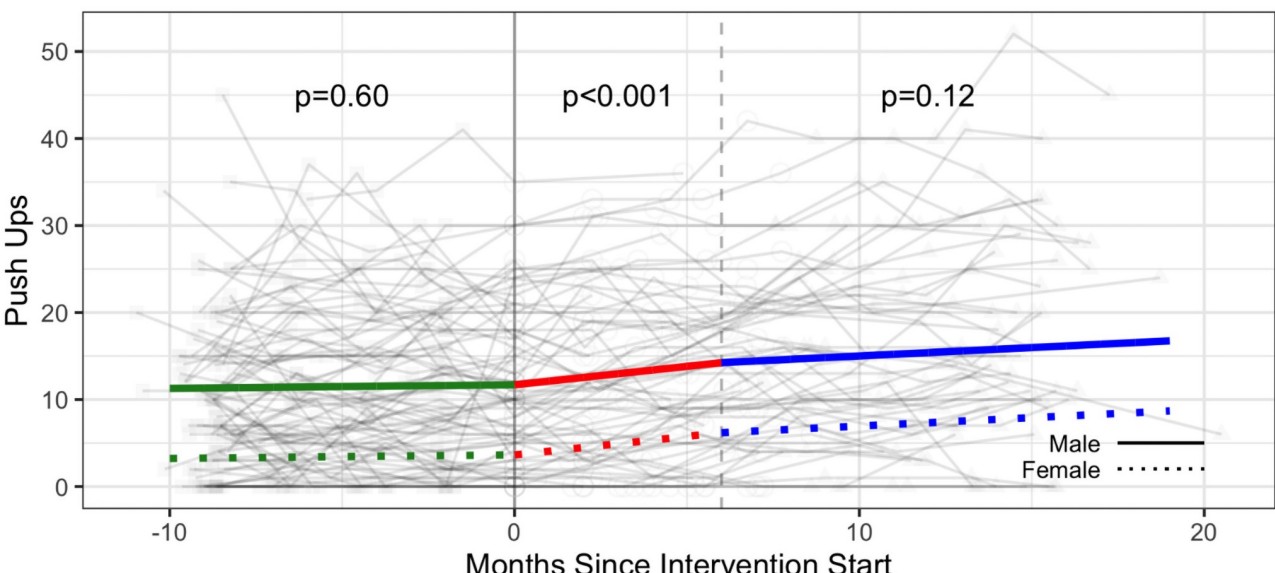

**Fig 6. Strength—Push ups (number of repetitions) trajectories over phase 1 (baseline), phase 2 (intervention) and phase 3 (follow-up monitoring); (n = 100 participants).** Vertical dotted line indicates end of phase 2 (intention to treat) at 6 months.

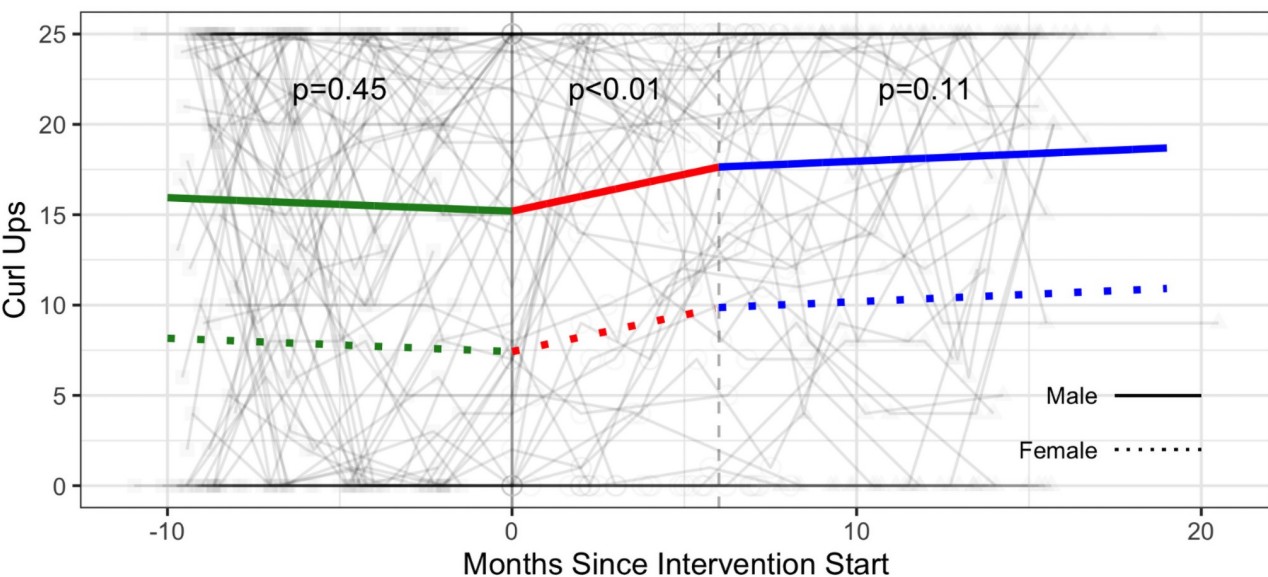

**Fig 7. Strength—Curl ups (number of repetitions up to 25 maximum) trajectories over phase 1 (baseline), phase 2 (intervention) and phase 3 (follow-up monitoring); (n = 98 participants).** Vertical dotted line indicates end of phase 2 (intention to treat) at 6 months.

*Per protocol analysis* (S5 Table). Of the 80 participants who started the intervention, 30 (38%) went over the intended 25 weeks, rescheduling their weekly coaching supervision up to 15 weeks into Phase 3 for a total of 40 weeks. Reasons for extending (or rescheduling) weekly coaching sessions included health reasons, travel, work commitments, change in coaching staff, or scheduling communication. When considering all coaching sessions attended by participants (Phase 2 and 3), the estimated intervention effect (taking Phase 1 into account)

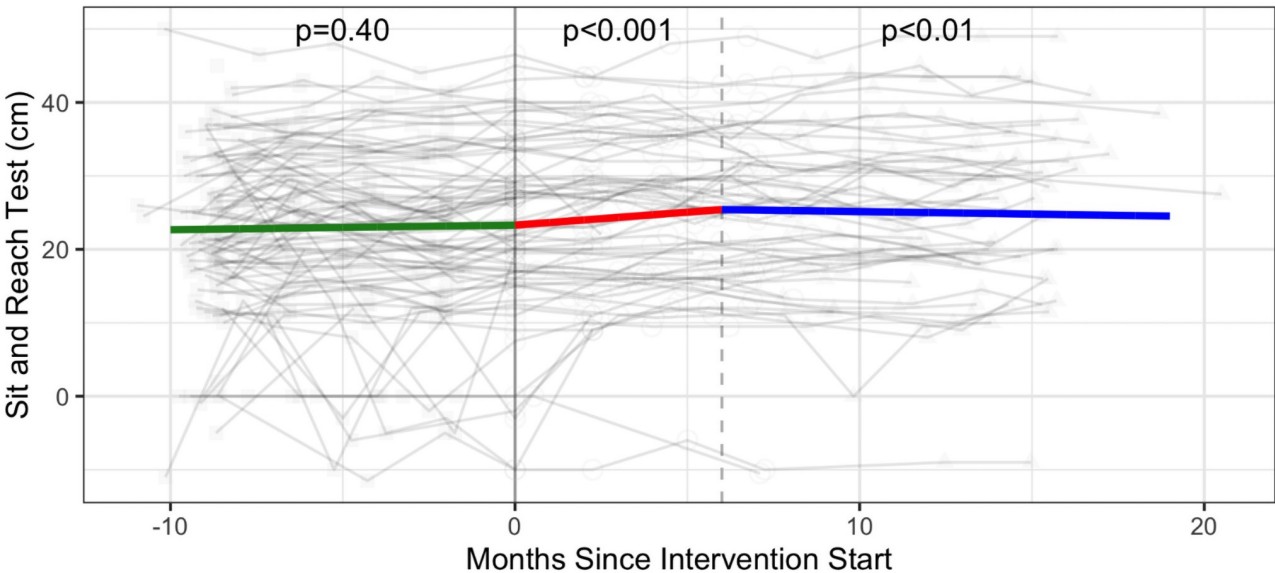

**Fig 8. Flexibility—Sit and reach test (cm) trajectories over phase 1 (baseline), phase 2 (intervention) and phase 3 (follow-up monitoring); (n = 99 participants).** Vertical dotted line indicates end of phase 2 (intention to treat) at 6 months.

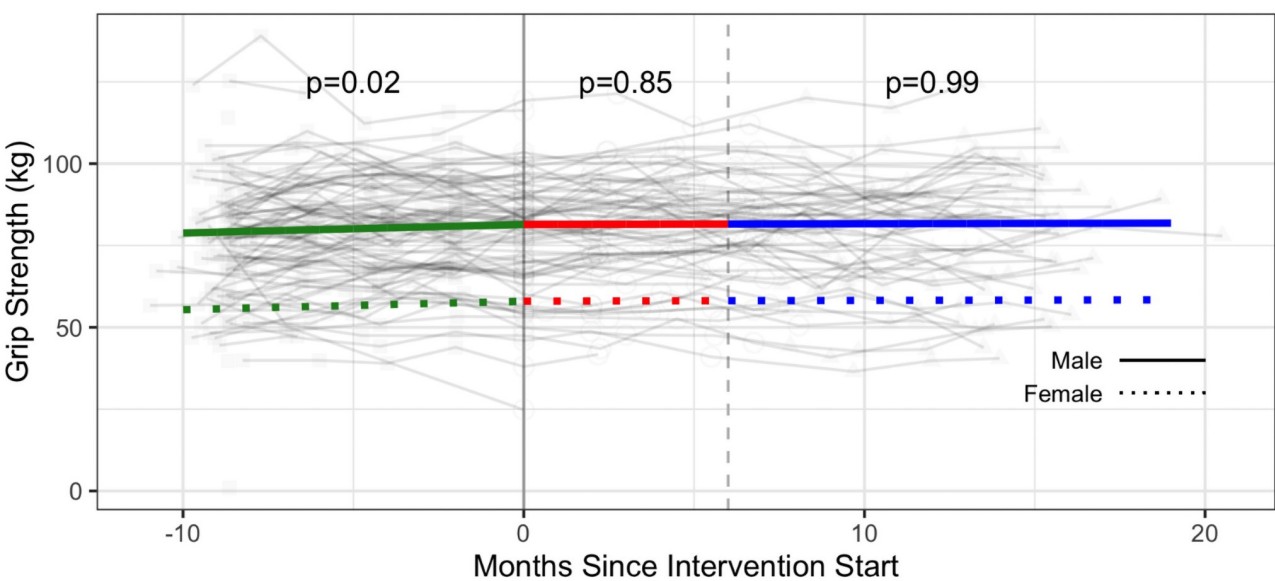

**Fig 9. Strength—Upper extremity—Grip strength (kg) trajectories over phase 1 (baseline), phase 2 (intervention) and phase 3 (follow-up monitoring); (n = 101 participants).** Vertical dotted line indicates end of phase 2 (intention to treat) at 6 months.

remained non significant for $\dot{V}O_2$peak (1.56mL/kg/min; 95% CI: -0.79, 3.90). (S5 Table details results of the per protocol analyses).

*Exercise dose* (S6 Table). Forty-five (45) participants completed at least one web-based weekly exercise questionnaire resulting in dose data included in the sub-group analysis examining the influence of dose on outcomes. The intervention by dose effect represents the expected increase in outcome for every additional minute spent exercising for every month in

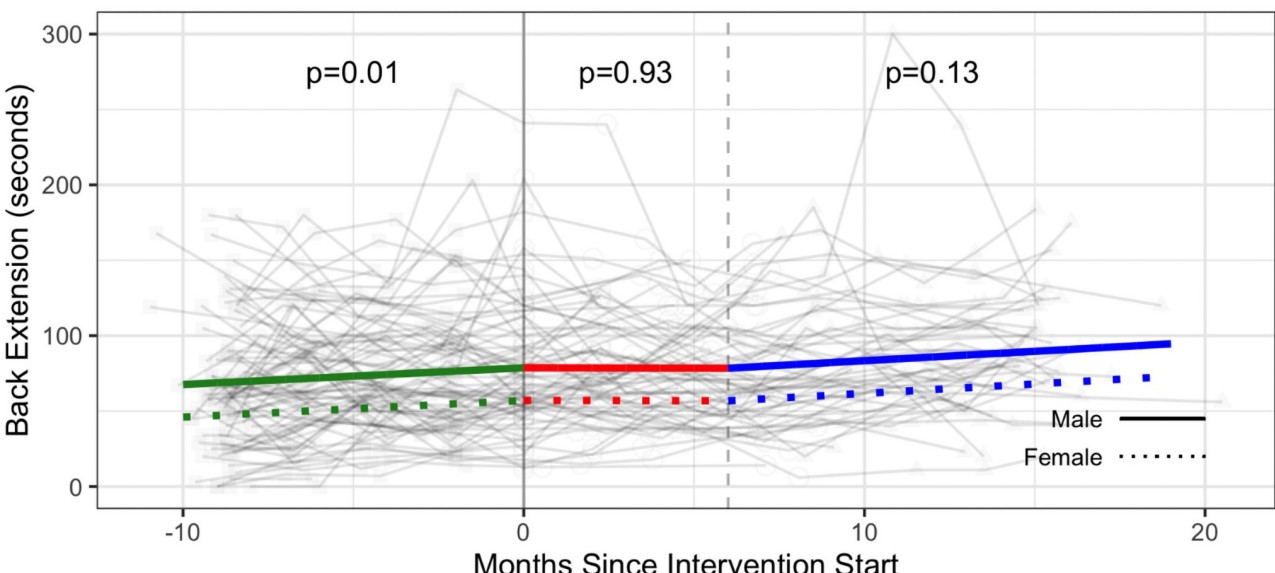

**Fig 10. Strength—Back extension (seconds) trajectories over phase 1 (baseline), phase 2 (intervention) and phase 3 (follow-up monitoring); (n = 97 participants).** Vertical dotted line indicates end of phase 2 (intention to treat) at 6 months.

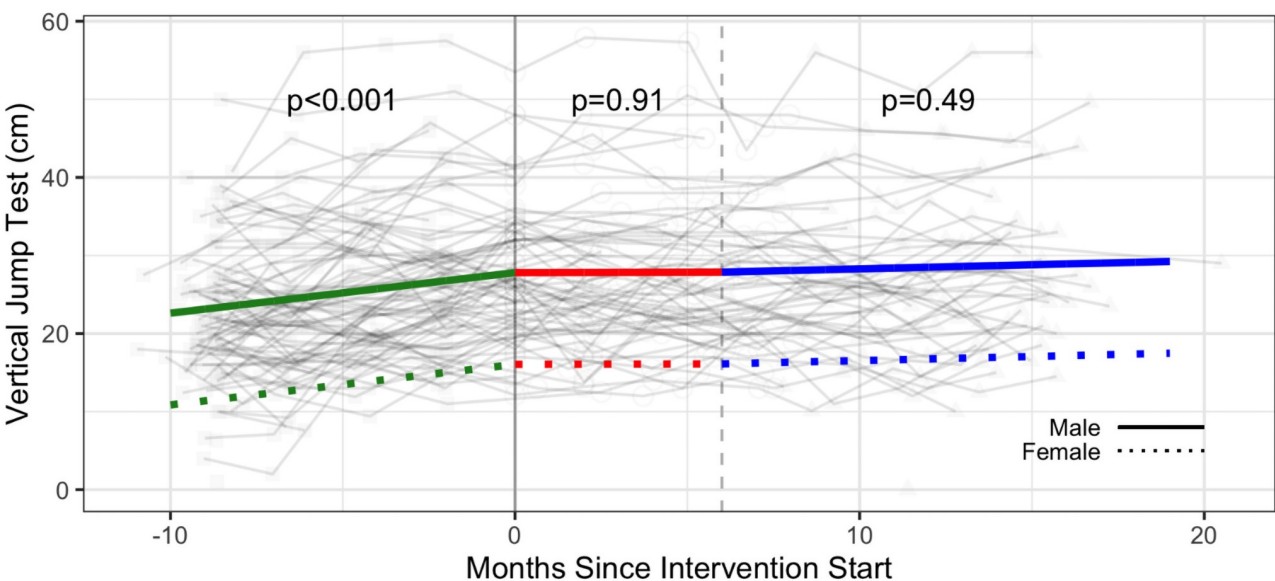

**Fig 11. Strength—Lower extremity—Vertical jump test (cm) trajectories over phase 1 (baseline), phase 2 (intervention) and phase 3 (follow-up monitoring); (n = 96 participants).** Vertical dotted line indicates end of phase 2 (intention to treat) at 6 months.

the study. Taking exercise dose into account, the primary outcome of $\dot{V}O_2$peak, remained non significant when considering exercise dose of $\geq$150 min/week of MVPA (-0.24 ml/kg/min; 95% CI: -1.48, 1.01) or $\geq$75 min/week of vigorous physical activity (-0.26 ml/kg/min; 95% CI: -1.68, 1.17) (S6 Table details the results of the exercise dose analysis).

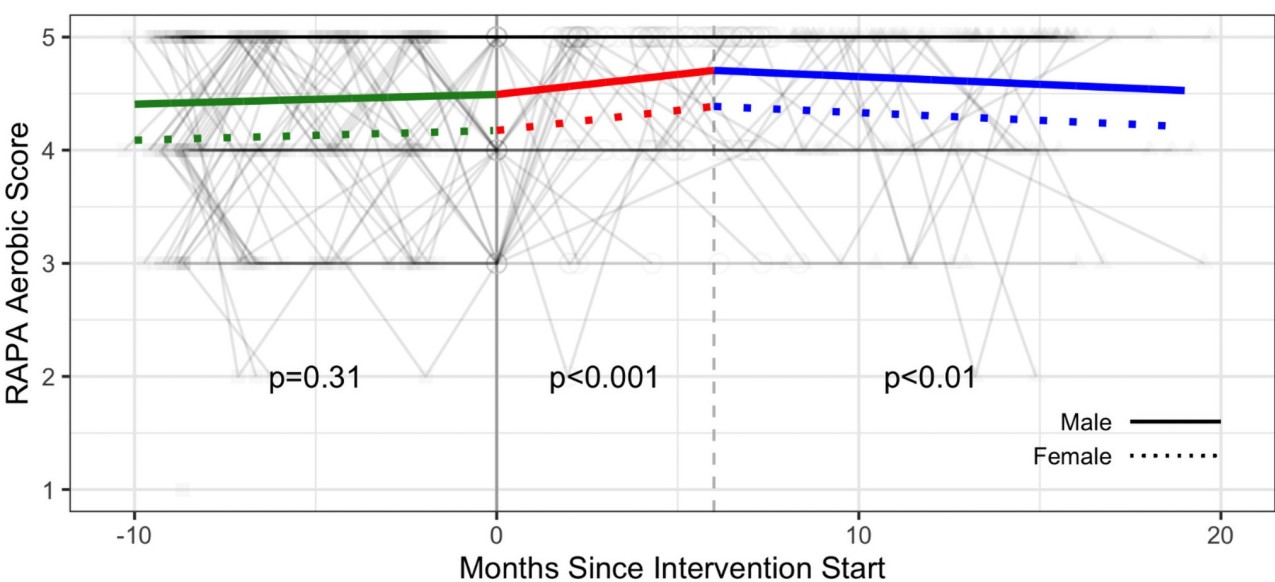

**Fig 12. Self-reported physical activity—Rapid Assessment of Physical Activity (RAPA) questionnaire—RAPA 1 (Aerobic) score trajectories over phase 1 (baseline), phase 2 (intervention) and phase 3 (follow-up monitoring); (n = 105 participants).** Vertical dotted line indicates end of phase 2 (intention to treat) at 6 months.

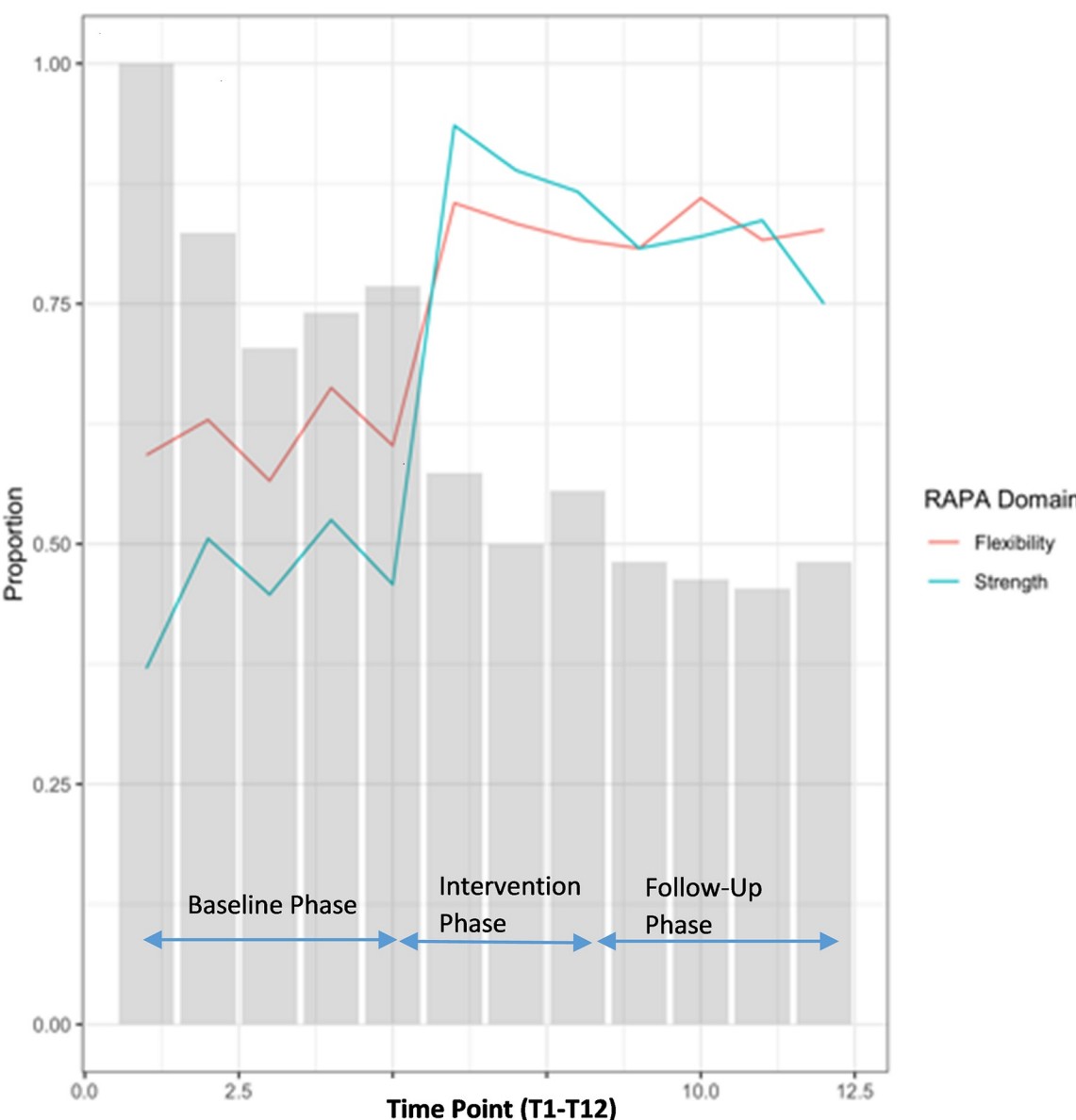

**Fig 13. Self-reported physical activity outcome—Rapid Assessment of Physical Activity (RAPA) questionnaire—RAPA 2 (Strength and flexibility) score trajectories over phase 1 (baseline), phase 2 (intervention) and phase 3 (follow-up monitoring); (n = 105 participants).**

## Discussion

In our study of adults living with HIV who engaged in this six-month CBE intervention, supervised weekly by a fitness coach, we observed no statistically or clinically significant changes in $\dot{V}O_2$peak (0.56ml/kg/min; 95% CI: -1.27, 2.39) after taking Phase 1 baseline into account.

Baseline $\dot{V}O_2$peak values in our study sample appear to be lower than reported in general populations. The mean baseline $\dot{V}O_2$peak among participants in this study (24.1 and 16.7 ml/kg/min in males and females, respectively) were lower than $\dot{V}O_2$max reported in the broader population (32-40ml/kg/min for males; 26-32ml/kg/min for females aged 40–59 years) [57].

Cardiorespiratory fitness among adults living with HIV is lower compared with the general population [58, 59] as previously reported by Vancampfort and colleagues who reported $\dot{V}O_2$max of 27.4ml/kg/min in a systematic review of 21 studies that pooled data from 1010 adults living with HIV (mean age 41 years; 64% males) [60]. Thus it is important to consider strategies to increase cardiorespiratory fitness among adults living with HIV.

Observed effects of the intervention on $\dot{V}O_2$peak in our study were inconclusive. While the point estimate of 0.56 ml/kg/min did not reach statistical significance, the lower and upper limits of the 95% CI (-1.27, 2.39) indicated both a deterioration and clinically important improvement in $\dot{V}O_2$peak with CBE. During the baseline phase we observed a small, non significant increase in $\dot{V}O_2$peak of 0.12 ml/kg/min per month (95% CI: -0.04, 0.28 ml/kg/min per month) followed by a statistically significant increase during the intervention at 0.21 ml/kg/min per month (95% CI: 0.01, 0.41 ml/kg/min per month), which corresponds to an expected increase of 1.26 ml/kg/min over a six-month intervention (Table 2). While the overall point estimate was <2 ml/kg/min (estimated threshold for important change) the upper limit of the confidence interval was compatible with a potential improvement of 2.46 ml/kg/min [13]. Nevertheless, the estimated effect of 1.26 ml/kg/min was lower than the systematic review literature that reported aerobic and resistance exercise performed at least thrice weekly for at least five weeks resulted in an increase in $\dot{V}O_2$max of 2.9 ml/kg/min among exercisers compared with non exercisers living with HIV [12, 13] and a meta-analysis (n = 10 studies) that reported significant improvements in $\dot{V}O_2$max of 3.8 ml/kg/min with at least four weeks of intervention among exercisers compared with non-exercisers [11]. These randomized controlled trials often involved highly supervised interventions (thrice weekly), performed in hospital-based rehabilitation settings opposed to our CBE intervention that involved only weekly supervision (Phase 2) followed by entirely independent exercise (Phase 3) in a community-based setting. Our study did not include a comparison group; instead we considered the Phase 1 baseline as a control or monitoring phase of the study from which to estimate an effect. We believe our estimated value of 0.56 ml/kg/min, the additional change during the intervention period, above that observed in the baseline phase, is a conservative estimate of the intervention effect because it provides a measure of the attributable effect of the CBE intervention beyond any effect of enrollment and monitoring in the study. If we interpreted no change in $\dot{V}O_2$peak occurring during Phase 1 baseline, the Phase 2 improvement in $\dot{V}O_2$peak would be estimated at 1.3 ml/kg/min, a greater improvement (0.211*6 months). While the point estimate remains below the estimated threshold for important change [13], ultimately the study lacks precision to determine either a clear benefit or harm with CBE for $\dot{V}O_2$peak. Furthermore, we measured $\dot{V}O_2$peak (not $\dot{V}O_2$max) which may explain lower values of cardiorespiratory fitness in our study. Similar non significant intervention effects of the point estimate after taking the baseline monitoring phase into account were found with self-reported aerobic physical activity, whereby despite a statistically significant monthly increase in RAPA 1 (aerobic) scores (+0.04 points/month) during the Phase 2 intervention, the overall improvement in aerobic physical activity across the 6 month intervention after taking the baseline phase into was non significant (0.16 points (95% CI: -0.03, 0.35) (S2 Table).

We hypothesize several reasons for our inconclusive findings related to $\dot{V}O_2$peak in our study that may include insufficient dose of exercise intensity and duration of intervention achieved by participants within the intervention [61]. The CBE intervention involved a combination of aerobic, resistance, balance and flexibility training, and was not specifically tailored to enhancing aerobic capacity. Oursler and colleagues found a significantly greater increase in $\dot{V}O_2$max of 3.6 ml/kg/min among people living with HIV who completed 16 weeks of vigorous intensity aerobic exercise compared with no change in $\dot{V}O_2$max among participants who completed 45 minutes of moderate-paced walking [10]. Erlandson and colleagues [62] conducted a

thrice weekly 24-week aerobic and resistance exercise intervention comparing older adults living with HIV and without HIV (50–75 years) where after 12 weeks of moderate-intensity of exercise individuals were randomized to 12 additional weeks of moderate or high intensity exercise. Despite similar $\dot{V}O_2$max at baseline, older adults with HIV had significantly greater improvements in $\dot{V}O_2$max, and physical function compared with older adults without HIV, and significantly greater improvements in strength among those randomized to high-intensity exercise [62]. In our study, we adopted a 'guided adaptation' approach whereby the intervention was tailored to the ongoing assessment of the needs and ability of participants by the fitness coach. Participants also were not compensated for their engagement in the CBE intervention as this would have introduced a co-intervention with incentive to exercise. This aligned with our implementation science approach, to evaluate the impact of the intervention in the 'real world' setting [34, 63]. There was also considerable variation of $\dot{V}O_2$peak within individuals over time which may indicate changes in fluctuations in health status that may occur among adults living with HIV [64] (S1 Fig). Given level of engagement can fluctuate based on level of readiness to exercise [65], along with health fluctuations for people living with HIV [21], it is unlikely all participants exercised at the prescribed intensity and duration.

Beneficial changes with the six-month CBE intervention were found for some secondary measures of cardiovascular health (systolic blood pressure), strength (push ups, curl ups), flexibility (sit and reach) and self-reported physical activity. We found a significant intervention effect with systolic blood pressure. The decreasing rate of systolic BP of ~1 mmHg / month (5 mmHg total) observed in the intervention phase of the study aligns with evidence that exercise can reduce or prevent hypertension among adults [66–68] including people living with HIV [69]. Our results align with systematic review evidence reporting benefits to systolic BP ranging from 1.8 to 10.9 mmHg with endurance and resistive training among adults [70]. Further systematic review evidence indicated a reduction in systolic BP of 10 mmHg can lower the risk of major cardiovascular disease events, coronary heart disease, stroke and heart failure lower risk of mortality from stroke [71]. Hence, results from this study may suggest a trend towards a potential clinically important benefit to cardiovascular health for people living with HIV [72]. Despite the decreasing trend in systolic BP observed in the intervention phase, results were ultimately inconclusive for $\dot{V}O_2$peak, highlighting this as an important area for investigation, examining the impact of CBE on cardiovascular health for people living with HIV.

Results demonstrated significant improvements for select strength (push ups and curl ups) and flexibility (sit and reach test) outcomes during the intervention after taking the baseline monitoring phase into account. Benefits to strength were sustained in the follow-up monitoring phase (Phase 3). We hypothesize reasons for improvements to strength and flexibility may be associated with specificity of training whereby push ups, curl ups, and hamstring flexibility assessed in the fitness testing were analogous to exercises in the participants' CBE intervention. Improvements in the baseline monitoring phase observed for some strength outcomes (grip strength, vertical jump, back extension) may reflect participants initiating independent exercise during the baseline phase or becoming familiarized with the strength assessments over time.

Improvements in some outcomes found during the intervention were not sustained in the follow-up phase, such as self-reported aerobic physical activity (Fig 12 and S2 Table). Similarly, a small decline was observed in $\dot{V}O_2$peak in Phase 3 (Table 2). We posit that weekly supervision from the fitness coaches during the Phase 2 intervention may be important for participants to remain engaged in the intervention (and the study). Results of a systematic review that investigated dropout from physical activity interventions suggest that supervised exercise with highly qualified personnel is associated with lower withdrawals from physical activity interventions among people living with HIV [73]. In this study, 84% (67/80) of participants

who started the Phase 2 intervention completed it 6 months later, and 77% (52/67) went on to complete the 8 month Phase 3 monitoring phase. Our intervention included a combination of features aimed at reducing barriers to engaging in exercise for people living with HIV such as providing a gym membership, personalized fitness instruction, and monthly self-management educational sessions; we also allowed participants to reschedule their weekly coaching sessions to accommodate for episodes of illness [21, 22]. Nevertheless, most benefits found during the intervention were not sustained in the follow-up monitoring phase, despite continued access to the YMCA membership, highlighting the potential importance of the one-on-one fitness coaching. Personalized coaching is a costly feature of exercise interventions whereby individuals may not be able to sustain out-of-pocket expenses for personalized training over the long term. Future research should explore the role of group-based exercise, peer-support and different self-management approaches as potentially less costly features of CBE interventions to promote the sustained uptake of physical activity over time [74, 75]. Ascertaining who might benefit most from supervised CBE, and identifying strategies to facilitate the sustained benefits of, and engagement in, physical activity after a CBE intervention needs further exploration.

To our knowledge, this is the first examination of cardiorespiratory fitness, cardiovascular health, strength, flexibility and physical activity involving a multi-phased CBE intervention in a 'real world' setting with adults living with HIV. Strengths include our prospective longitudinal three-phased study design whereby data were collected at multiple timepoints before and after CBE introduction at T5 to detect whether CBE had an effect significantly greater than the underlying baseline trend [76]. Additional strengths included our implementation science approach offering insights on changes in outcomes that can occur among adults living with HIV exercising in a 'real world' community fitness setting, and the sustainability of the intervention-effect over time with an independent self-maintenance phase of exercise. By examining the impact of this six-month supervised CBE intervention we focused on the 'E-effectiveness' component of the RE-AIM framework within our broader implementation science approach [42]. In other work, we report on subsequent features of adoption and implementation to further understand the process, uptake and sustainability of implementing CBE with adults living with HIV in the community [77].

## Limitations

We did not achieve our original intended sample size of 75 participants at study completion due to participant withdrawal, and particpants may have missed assessments across the 22-month phase of study. However, we used mixed models including available data for all participants across the three phases. Because of the difficulty in adjusting for multiple comparisons with correlated outcomes we did not adjust our p-values, but have presented confidence intervals for our estimates. We did not adjust for multiple comparisons of the secondary outcomes [78]. As such, results should be considered exploratory. We did not objectively measure exercise intensity of each session, and using weekly self-report recall which limited our ability to measure the intensity of exercise among participants in the study. Finally few females (<15% of the sample) participated in the study. Future work should consider CBE features to address structural, geographical and personal barriers to engaging in exercise [79, 80]. Lastly, this study was conducted in a single urban community fitness centre in Toronto, Canada. Participants who enrolled in this 22-month study may have been more likely to possess a higher level of self-health awareness compared to the broader HIV population. Generalizability of findings to the broader HIV population, other fitness centres, and in other contexts such as low-and middle income countries is unclear.

## Conclusions

Community-dwelling adults living with HIV who engaged in a six-month CBE intervention supervised weekly by a fitness coach demonstrated inconclusive results with respect to $\dot{V}O_2$peak, and potential improvements in some outcomes of cardiovascular health, strength, flexibility and self-reported physical activity. CBE is an approach to consider for improving health outcomes with adults living with HIV within a self-management framework. Future research should consider features tailored to promote uptake and sustained engagement in independent exercise among adults living with episodic illness such as HIV.

## Supporting information

**S1 Fig. Individual $\dot{V}O_2$peak trajectory plots across Phase 1 (baseline), Phase 2 (intervention) and Phase 3 (follow-up monitoring).** The line indicates participant-level predictions from the model. Trajectories indicate the fitted values across all three study phases. Red dots indicate $\dot{V}O_2$peak measurements in Phase 1 (baseline monitoring). Green dots indicate $\dot{V}O_2$peak measurements in Phase 2 (intervention). Blue dots indicate $\dot{V}O_2$peak measurements in Phase 3 (follow-up monitoring).
(PDF)

**S1 File. Statistical supplement.**
(PDF)

**S2 File. Research ethics board approval letter and original approved protocol (April 2016).**
(PDF)

**S1 Table. Characteristics of participants at study initiation in the study sample at enrollment, start of intervention, end of intervention, and end of follow-up (end of study).** Characteristics for participants who completed demographic questionnaire at study enrollment (Time 1; Month 0): [a] excluding HIV.
(PDF)

**S2 Table. Results—Secondary outcomes—Primary (Intention to treat) analysis.** Slopes are change in outcome over one month. [a] Significant trend in Phase 1 (baseline) slope versus 0; [b] Significant trend in Phase 2 (intervention) slope versus 0; [c] Significant change in Phase 3 (follow-up monitoring) slope versus Phase 2 (intervention) slope. [d] Follow-up slope is the difference in slope between the follow-up and intervention phase; RAPA: Rapid Assessment of Physical Activity Questionnaire (higher scores indicate greater physical activity).
(PDF)

**S3 Table. RAPA 2—Strength and flexibility scores.** [a]One of these participants did not have T8 data so is not included in McNemar's below; [b]For strength counts, participants reporting either Strength or Both were combined; [c] For flexibility counts, participants reporting either Flexibility or Both were combined; [d] Same number of participants moving from not engaging to engaging in flexibility were the same as the number of participants moving from engaging in flexibility to not; RAPA: Rapid Assessment of Physical Activity Questionnaire.
(PDF)

**S4 Table. Overall estimated effects of the Phase 2—CBE intervention after taking the Phase 1—Baseline monitoring into account for all outcomes and analyses: Primary analysis (intention to treat) and secondary (exploratory, post hoc) analyses (per protocol and exercise dose analysis).** [a] Significant overall estimated treatment effect over the six-month intervention after taking the baseline monitoring into account; RAPA: Rapid Assessment of

Physical Activity; CI: Confidence Interval; V̇O2peak: peak oxygen consumption. *Sample size*: Intention to Treat Analysis: Maximum n = 105 participants; Post Hoc Exploratory Per Protocol Analysis: Maximum n = 80 participants; Post Hoc Exploratory Dosage Analysis: Models A and B: Maximum: 45 participants.
(PDF)

**S5 Table. Post hoc exploratory analyses—Per protocol—Comparing trends during the baseline monitoring phase (defined as any time prior to the first supervised training session) and the trends while receiving the intervention (defined as the time between the first and last supervised training session).** Slopes are change in outcome over one month; Baseline slope (trend) p value <0.05 = significant difference in baseline slope versus 0; Slope while receiving the intervention (trend) p value <0.05 = significant difference in slope during the intervention versus 0; RAPA: Rapid Assessment of Physical Activity; RAPA Aerobic (range: 1–5); CI: Confidence Interval.
(PDF)

**S6 Table. Post hoc exploratory analyses—Exercise dose interaction of exercise dose and duration of the intervention on all outcomes.** Model A: Dose defined as total number of Moderate + Vigorous minutes of activity in the past week; Model B: Dose defined as total number of Vigorous minutes of activity in the past week. Canadian Physical Activity Guidelines (CPAG): Model A: ≥150 minutes moderate to vigorous / week and Model B: ≥75 minutes vigorous activity. Slopes are change in outcome over one month. [a]Follow-up slope is the difference in slope between the follow-up and intervention phase; Phase 1: Baseline slope p value <0.05: statistically significant difference in Phase 1 (baseline) slope versus 0; Phase 2: Intervention slope p value <0.05: statistically significant difference in Phase 2 (intervention) slope versus 0; Phase 3: Follow-up slope p value <0.05: statistically significant change in Phase 3 (follow-up monitoring) slope versus Phase 2 (intervention) slope.
(PDF)

**S1 Checklist. TREND checklist.**
(PDF)

## Acknowledgments

We acknowledge the adults living with HIV who participated in this community-based exercise study.

We acknowledge the Central Toronto YMCA staff for their longstanding collaboration in this study, including Mehdi Zobeiry, Ivan Ilic, Zoran Pandovski, Shane Stanford, Karina Gomez, Dwayne Campbell, Camila Paino, Abi Shan, Sherif El Abdul, Kanako Isobe, Tommy Berrospi, Nataliya Zlotnikov, Dom Hanlon, Cristina Granados, Letizia Lepore, Gaya Navaratnam, Derek Quick, Katie Lowe, Dexter Wilson, Liam Dick, Elfi Vinagre, Katherine Foster Grajewski, Emily Sas, Katharine Stanbridge, Christine Hsu, Katie Lowe, and Maria Rapalini.

We thank the following community-based organizations and individuals for their expertise, ongoing guidance, involvement and support with this work: Realize (Kate Murzin, Tammy Yates), Toronto PWA Foundation (Chris Godi), AIDS Committee of Toronto, and Casey House (Soo Chan Carusone), Greg Robinson, Ken King, and James Murray.

### Dedication

We dedicate this manuscript to Shane Stanford, YMCA staff and dedicated member of the CBE study fitness training team.

## Author Contributions

**Conceptualization:** Kelly K. O'Brien, Aileen M. Davis, Soo Chan Carusone, Ada Tang, Patricia Solomon, Mehdi Zobeiry, Ivan Ilic, Ahmed M. Bayoumi.

**Data curation:** Kelly K. O'Brien, Ivan Ilic.

**Formal analysis:** Kelly K. O'Brien, Aileen M. Davis, Soo Chan Carusone, Lisa Avery, Ada Tang, Rachel Aubry, Ahmed M. Bayoumi.

**Funding acquisition:** Kelly K. O'Brien, Aileen M. Davis, Soo Chan Carusone, Ada Tang, Patricia Solomon, Mehdi Zobeiry, Ahmed M. Bayoumi.

**Investigation:** Kelly K. O'Brien, Aileen M. Davis, Soo Chan Carusone, Ada Tang, Patricia Solomon, Rachel Aubry, Ivan Ilic, Zoran Pandovski, Ahmed M. Bayoumi.

**Methodology:** Kelly K. O'Brien, Aileen M. Davis, Soo Chan Carusone, Lisa Avery, Ada Tang, Patricia Solomon, Mehdi Zobeiry, Ivan Ilic, Zoran Pandovski, Ahmed M. Bayoumi.

**Project administration:** Kelly K. O'Brien, Rachel Aubry, Mehdi Zobeiry, Ivan Ilic, Zoran Pandovski.

**Supervision:** Kelly K. O'Brien.

**Validation:** Kelly K. O'Brien, Aileen M. Davis, Soo Chan Carusone, Lisa Avery, Ada Tang, Patricia Solomon, Rachel Aubry, Ivan Ilic, Ahmed M. Bayoumi.

**Writing – original draft:** Kelly K. O'Brien, Aileen M. Davis, Soo Chan Carusone, Lisa Avery, Rachel Aubry, Ahmed M. Bayoumi.

**Writing – review & editing:** Kelly K. O'Brien, Aileen M. Davis, Soo Chan Carusone, Lisa Avery, Ada Tang, Patricia Solomon, Rachel Aubry, Mehdi Zobeiry, Ivan Ilic, Zoran Pandovski, Ahmed M. Bayoumi.

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
