## [Decision Letter · Decision Letter 0]

7 Jul 2021

PONE-D-21-15069

Examining the impact of a community-based exercise (CBE) intervention on cardiorespiratory fitness, cardiovascular health, strength, flexibility and physical activity among adults living with HIV: an interrupted time series study

PLOS ONE

Dear Dr. O'Brien,

Thank you for submitting your manuscript to PLOS ONE. After careful consideration, we feel that it has merit but does not fully meet PLOS ONE’s publication criteria as it currently stands. Therefore, we invite you to submit a revised version of the manuscript that addresses the points raised during the review process.

We look forward to receiving your revised manuscript.

Kind regards,

Walid Kamal Abdelbasset, Ph.D.

Academic Editor

PLOS ONE

Journal Requirements:

2. Please state in your methods section the date range when this study took place.

3. Please ensure that all deviations from the original approved protocol are discussed and justified in the manuscript text. For instance we have noted that one of the primary outcomes reported in the protocols is disability as measured by the HIV Disability Questionnaire (HDQ). However this measurement is not reported in the manuscript text. Please provide some further clarification on this.

4. Please ensure you have included the registration number for the clinical trial in the manuscript.

Reviewers' comments:

Reviewer's Responses to Questions

**Comments to the Author**

1. Is the manuscript technically sound, and do the data support the conclusions?

Reviewer #1: Yes

Reviewer #2: Partly

Reviewer #3: Yes

2. Has the statistical analysis been performed appropriately and rigorously? 

Reviewer #1: Yes

Reviewer #2: No

Reviewer #3: Yes

3. Have the authors made all data underlying the findings in their manuscript fully available?

Reviewer #1: No

Reviewer #2: Yes

Reviewer #3: Yes

4. Is the manuscript presented in an intelligible fashion and written in standard English?

Reviewer #1: Yes

Reviewer #2: Yes

Reviewer #3: Yes

5. Review Comments to the Author

Reviewer #1: General Comments: I truly enjoyed reading about your study and feel that it explores a very relevant issue. I worked in HIV/AIDS services in the early 1990s in New York City and to see this sample of HIV+ individuals living relatively healthy lives, and exercising, and making it work is pretty indescribable to me. It is what the PWAs and caregivers all hoped and prayed for in those dark days and honestly never believed would happen. I just wanted to tell you my personal perspective and commend you on your work with your participants and clients.

I think the study should be published with some major rewriting related, honestly, to increasing the clarity of your presentation of your methodology and description of your statistics and results. It got a bit muddled at times but I feel that this could fairly easily be rectified.

Minor Changes

-In title don’t use abbreviations (CBE)

-Line 55 Merge this paragraph with prior one sentence paragraph

-Lines 69 to 70 -In abstract results, you should state that V02 peak change was not significant. You list the values, but it is confusing and we don’t find out that it wasn’t significant until later.

-I could not find whether participants were compensated for this? Did I miss that? If not, I wonder if it would have helped participation and statistical significance? It is a big study and I am surprised that compensation was not built in.

-Line 126 Please state credentials of the fitness coaches used in study

-Between ref 37 and 38 there is an extra non-numbered reference that needs to be fixed.

Major Changes

- Lines 127-128.

I would like to see a short explanation of why you chose your secondary measures, You could have gone in a lot of different directions on this.

-Lines 129. An explanation of why you chose the time periods of 8 mos, 6 mos and 8 mos. Would be helpful

-Lines 213-214 The lack of blinding is buried here and needs to be stated at the start of the methodology section

-Lines 226 to 235 Please provide a rationale for the design of this Exercise Dose questionnaire. Was it adapted from an established validated instrument? Similarly the demographics questionnaire

-Line 237 Change title to Statistical Analysis. Your stats are somewhat complex but this section is a bit long and it seems like you are mixing the description of how you are measuring your variables with the statistical tests you will use. Some editing to make this tighter and easier to understand for the reader would be good..

-Lines 238 to 241 I would delete this paragraph. It adds little to the explanation of your analysis.

-Line 363 Thoughts on why the decline in V02 max in phase 3 – wasn’t this somewhat expected?

-Table 3. It would be so much clearer if you would add 2 columns that listed the actual values for HR and BP before and after. You are making the reader work too hard to understand your results.

-Line 408 to 411 Could you please state whether the strength and flexibility improvements are significant.

-Line 470 You begin to refer to S5 and S7 here but I can’t find where you previously defined these terms. I see later that it refers to the Supplements. Please provide a little explanation here.

-Limitations: Wasn’t your internal validity affected by the wide variation of the types of exercise participants engaged in? IE group vs individual, spinning vs yoga. Did you control statistically for this?

Reviewer #2: This is an interesting although dated (enrolment ended in January 2017) real-world evaluation of a community-based exercise (CBE) intervention cardiorespiratory fitness, cardiovascular health, strength, flexibility and physical activity among adults living with HIV seen for care in Toronto Canada.

I have a few remarks mainly about the statistical analysis and interpretation of the results.

Main points

1. The main conclusion seems to be that CBE had no effect on the primary outcome of change in peak VO2. I respectfully disagree with the authors on this point. The estimated change in peak V̇O2 attributed to the six-month phase 2 CBE intervention was 0.56 ml/kg/min (95% Confidence Interval (CI): -1.27, 2.39). Thus although p-value was >0.05 the data are compatible with a 2.39 ml/kg/min increase in peak VO2, which according to the authors is clinically significant (>2.0 ml/kg/min). Therefore, the study is inconclusive as both possible harm and benefit of CBE cannot be ruled out because of low statistical power. Suggest to rephrase text and conclusions accordingly.

2. The distinction between statistical and clinical significance applies also for the secondary endpoints. Thus, for example, what are the possible clinical implications of a reduction of 5.18 mmHg in systolic blood pressure? How does this translate into a reduction of risk of cardiovascular morbidity and mortality? Authors should consider to estimate these risks by means of the Framingham or D.A.D CVD risk scores. Statistical significance without clinical significance has little relevance.

3. One key factor that might explain the discrepancies with the results of other studies is the particular case-mix of the target population. Although participants had to be ‘medically stable and safe to participate in exercise activities’ Table 1 indicates that >35% had muscle pain or arthritis at baseline, 19% reported bad health and 82% had at least two other co-morbidities. This issue seems to be confirmed by the average level of peak VO2 at baseline. Is it conceivable that if peak VO2 drops below a certain level CBE is bound to be less effective? Authors should try a sensitivity analysis after excluding the n=20 patients who self-reported bad health at enrolment?

4. Mixed linear model assumes that the outcome has a symmetric (Normal) distribution. Unclear whether this was the case for all endpoints, or whether scale transformations might be needed. Some of the secondary endpoints are discrete scores obtained from a questionnaire (e.g. RAPA, ranging from 1 to 5). Maybe a Poisson model would have been more realistic distribution assumption for these?

5. I do not follow the hybrid parametrization which was used in the mixed linear model regarding the change of slope over time. The two most commonly used parametrisations are i) the beta coefficients are the estimates of the slope in certain time window or ii) the betas are the estimates of the change in slope in those same windows. The authors appeared to have used an hybrid of these with the betas for the phase 1 and phase 2 following parametrisation i) while the betas for phase 3 following parametrisation ii). I think this is confusing for the reader. I believe the important contrasts are the slope of phase 2 vs. phase 1 and the slope of phase 3 vs. phase 2. Corresponding correct p-values should be given also in Figure 2.

6. There is substantial attrition with only 84% completing phase 2 and 77% phase 3. Missing data can introduce bias in mixed linear model analysis. The mentioned paper by Peter et al is a specific scenario in which only peak VO2 is simulated to be missing. If a person drops out from the study, all other relevant covariates will be also missing, starting from the exposure. In this alternative scenario where the imputation model would account for relevant predictor variables that are not incorporated in the mixed model multiple imputation is instead beneficial (see paragraph 4.4 Generalizability in Peter et al). So something better needs to be done to handle missing data.

7. I would remove the ‘an interrupted time series (ITS)’ description in the title and elsewhere (e.g. abstract line 120, 608). Again I think that could be misleading as ARIMA models are typically used for time series data. However, a different methodology has been implemented here.

8. The study has no control group. I assume that peak VO2 could have changed over time (from phase 1 to phase 2) as effect of simple regression to the mean. How was this effect accounted for in the analysis?

9. Sampling was conducted on voluntary basis. This could have led to have a selected sample of people with higher self-health awareness, level of education (38% had at least a University degree), etc.. Do the authors have characteristics of the whole YMCA population for comparison? Because these factors are also a possible cause of cardiovascular disease, the possibility of having introduced collider bias should be better discussed.

Other points

1. Line 616. RE-AIM framework is introduced here for the first time. Should be spelled out and better described

2. Lines 272-274. Unclear sentence. If removing sex resulted in a reduced model fit than sex should be retained in the model, not removed? Typo ‘adust’ instead of adjust

3. Line 301. Although it is explained in detail in Supplementary material, what was the minimum improvement in peak VO2 that the authors did not want to miss when calculating the sample size (2.39 ml/kg/min?). This needs to be clearly stated here instead of the generic ‘a trend change’.

4. Lines 316-318. Why were minutes of Self-Reported Physical Activity not collected for phase 1? Seem important baseline data given the study design.

5. Page 19. Table 1. I would report in the Table itself that the slopes estimate refer to changes per month. Regarding the legend. aFollow-up slope is the difference in slope between the intervention and baseline phase; this part seems to be incorrect and it is not matching the text in line 364; the slope in phase 3 should be calculated as the sum of the slope in phase 2 (+0.21) and the change from phase 2 vs. phase 3 (-0.37) =-0.16 ml/kg/min/month. I would report this in the table and not -0.37 to improve clarity ( see main point #5 above).

6. Line 484. I assume this is referring to the interaction term (duration of treatment phase multiplied by dose). Please rephrase, avoiding statistical jargon

7. Line 526-529. I do not follow, if anything the possible effect of the regression to the mean needs to be accounted for?

8. Lines 626-627. The authors make a big deal of the differences in secondary outcomes based on weak evidence (p-value non controlled for multiple comparisons) in Conclusions, Abstract, etc. So this sentence seems contradictory.

9. Lines 632-634. There could be a more general issue of selection bias (see main points #3 and #9 above) and possible implications for internal validity are unclear.

Reviewer #3: The present study shown in this paper had done a perfect job on reporting a long-term tracking clinical trials by applying exercise intervention on HIV patients. The methods utilited here was appropriate, the results were correct, and the conclusion they finally gain was reliable. At last but at least, they reported that the effect of exercise intervention on HIV therapy was effective, which enlighted patients with hopeful treatment stratgy. And again, as they metioned in the ending, featured exercise thrapeutic regimen needed to be elucidated.

Based on its topic and the perfect work they have done, I strongly recommend this paper to be publised in PLOS ONE.

6. PLOS authors have the option to publish the peer review history of their article (what does this mean?). If published, this will include your full peer review and any attached files.

Reviewer #1: No

Reviewer #2: No

Reviewer #3: **Yes: **Di Cui

---

## [Author Response · Author response to Decision Letter 0]

16 Aug 2021

**see attached response to reviewers attached with the files (similar to response below) - August 16, 2021**

PONE-D-21-15069 - Examining the impact of a community-based exercise intervention on cardiorespiratory fitness, cardiovascular health, strength, flexibility and physical activity among adults living with HIV: a three-phased intervention study

Dear Dr. Abdelbasset,

Thank you for the reviews of our above named manuscript. We appreciate you and the reviewers taking the time to review of our manuscript. We uploaded a revised track change version of the manuscript with revisions highlighted in yellow along with an unmarked version of the revised manuscript. Our point-by-point response to the reviewers’ comments is below. 

Declaration of Conflicting Interests: The author(s) declared no potential conflicts of interest with respect to the research, authorship, and/or publication of this article.

Funding: The author(s) disclosed receipt of the following financial support for the research, authorship, and/or publication of this article: This research is funded by the Canadian Institutes of Health Research (CIHR) HIV/AIDS Community-Based Research Initiative CIHR Funding Reference Number: 139685. Kelly K. O’Brien is supported by a Canada Research Chair in Episodic Disability and Rehabilitation from the Canada Research Chairs Program.

Please do not hesitate to contact me if you require any further information. Thank you for considering our manuscript for publication in PLOS ONE.

Sincerely,

Kelly O’Brien, PhD, PT

Associate Professor

Canada Research Chair in Episodic Disability and Rehabilitation

Department of Physical Therapy

500 University Avenue, Room 160

University of Toronto

kelly.obrien@utoronto.ca

Journal Requirements:

1. Please ensure that your manuscript meets PLOS ONE's style requirements, including those for file naming. The PLOS ONE style templates can be found at: https://journals.plos.org/plosone/s/file?id=wjVg/PLOSOne_formatting_sample_main_body.pdf and

Response: We revised the file naming according to the requested format and pared down the word count of the abstract. 

2. Please state in your methods section the date range when this study took place.

Response: The study dates are similar to the dates for data collection. These are included in the data collection section (Line 206). We added them again to the beginning of the methods (Line 133). The study dates are also included in Figure 1 – Participant Flow Chart.

3. Please ensure that all deviations from the original approved protocol are discussed and justified in the manuscript text. For instance we have noted that one of the primary outcomes reported in the protocols is disability as measured by the HIV Disability Questionnaire (HDQ). However this measurement is not reported in the manuscript text. Please provide some further clarification on this.

Response: Our study included primary and secondary outcomes of interest measured by objective and self-reported questionnaire assessments. This manuscript reports on the primary outcome of interest (VO2peak) and objectives and self-reported physical health assessments related to the primary outcome of interest. The HIV Disability Questionnaire was a secondary outcome of this study, beyond the scope of this manuscript. We added this detail to the manuscript (Line 206-209).

4. Please ensure you have included the registration number for the clinical trial in the manuscript.

Response: We added the clinical trial registration number to the manuscript (Line 148). It is also on the title page. ClinicalTrials.gov Identifier: NCT02794415 https://clinicaltrials.gov/ct2/show/record/NCT02794415 (Line 42-43).

Response: We revised so that the two sections are similar. 

Declaration of Conflicting Interests: The author(s) declared no potential conflicts of interest with respect to the research, authorship, and/or publication of this article. 

Funding: The author(s) disclosed receipt of the following financial support for the research, authorship, and/or publication of this article: This research is funded by the Canadian Institutes of Health Research (CIHR) HIV/AIDS Community-Based Research Initiative CIHR Funding Reference Number (FRN): CBR-139685. Kelly K. O’Brien is supported by a Canada Research Chair in Episodic Disability and Rehabilitation from the Canada Research Chairs Program.

Reviewer Comments

Reviewer #1: 

1. General Comments: I truly enjoyed reading about your study and feel that it explores a very relevant issue. I worked in HIV/AIDS services in the early 1990s in New York City and to see this sample of HIV+ individuals living relatively healthy lives, and exercising, and making it work is pretty indescribable to me. It is what the PWAs and caregivers all hoped and prayed for in those dark days and honestly never believed would happen. I just wanted to tell you my personal perspective and commend you on your work with your participants and clients.

Response: Thank you

2. I think the study should be published with some major rewriting related, honestly, to increasing the clarity of your presentation of your methodology and description of your statistics and results. It got a bit muddled at times but I feel that this could fairly easily be rectified.

Minor Changes

-In title don’t use abbreviations (CBE)

Response: We removed the CBE abbreviation from the title.

3. Line 55 Merge this paragraph with prior one sentence paragraph

Lines 69 to 70 -In abstract results, you should state that V02 peak change was not significant. You list the values, but it is confusing and we don’t find out that it wasn’t significant until later.

Response: We revised the abstract results in relation to reviewer #2 comments stating that VO2peak results were inconclusive (as reflected by the confidence interval indicating both an increase and decrease of Vo2peak with CBE) (Line 88).

4. I could not find whether participants were compensated for this? Did I miss that? If not, I wonder if it would have helped participation and statistical significance? It is a big study and I am surprised that compensation was not built in.

Response: Participants were not compensated for their engagement in the CBE intervention as this would have introduced a co-intervention with incentive to exercise. Participants received weekly personal training / supervision during Phase 2 (intervention) as well as a YMCA membership providing access to the YMCA throughout Phase 2 and Phase 3. Details of receiving a YMCA membership and access to the fitness coach is included in the description of the Intervention. (Line 172-174; Line 197-198).

5. Line 126 Please state credentials of the fitness coaches used in study

Response: All coaches were certified in personal training (Canada YMCA Personal Trainer Certification, CAN FIT PRO Personal Training Specialist). We added the certification details in the methods (Intervention section). (Line 173-174)

6. Between ref 37 and 38 there is an extra non-numbered reference that needs to be fixed.

Response: Thank you – we fixed the formatting of this reference.

Major Changes

7. Lines 127-128. I would like to see a short explanation of why you chose your secondary measures, You could have gone in a lot of different directions on this.

Response: Our study included primary and secondary outcomes of interest measured by objective and self-reported questionnaire assessments. This study reports on the primary outcome of interest (Vo2peak) and self-reported and objective physical health assessments related to the primary outcome of interest. The secondary outcomes focused on objective measures of cardiopulmonary fitness, strength, and flexibility, as they are related to the primary outcome of interest, Vo2peak, and commonly reported in previous systematic reviews on HIV and exercise. (Line 207-209)

8. Lines 129. An explanation of why you chose the time periods of 8 mos, 6 mos and 8 mos. Would be helpful

Response: We chose a 6-month CBE duration because it aligns with transtheoretical model evidence on the stages of behaviour change whereby the ‘action’ of practicing a new behaviour (exercise) lasts up to 24 weeks followed by the ‘maintenance’ stage during which commitment to sustaining the new behaviour (self-monitored exercise) is solidified (Prochaska, 1997). The 8 month baseline and 8 month follow-ensured equal timeframes prior to and after the intervention, ensured an equal number of data points in each phase (Baseline: T1-T4; Intervention: T5-T8; Follow-Up: T9-12). We include these details in our protocol referenced in the manuscript. 

9. Lines 213-214 The lack of blinding is buried here and needs to be stated at the start of the methodology section 

Response: We added a statement at the beginning of the description of the intervention specifying that participants were not blinded to the intervention. We retained the sentence about blinding of assessors in the data collection section (Line 170-171).

10. -Lines 226 to 235 Please provide a rationale for the design of this Exercise Dose questionnaire. Was it adapted from an established validated instrument? Similarly the demographics questionnaire

Response: Our questions about dosage in the demographic questionnaire were guided by the Canadian Physical Activity Guidelines (CPAG), which uses number of minutes engaged in moderate to vigorous aerobic physical activity in the past week as a unit of measurement. This was not from an existing validated instrument. The demographic questionnaire also was not a previous validated instrument. As a team, we identified constructs important for personal, health-related and exercise-related characteristics. We piloted the questionnaire prior to implementation with members of the team.

11. Line 237 Change title to Statistical Analysis. Your stats are somewhat complex but this section is a bit long and it seems like you are mixing the description of how you are measuring your variables with the statistical tests you will use. Some editing to make this tighter and easier to understand for the reader would be good.

Response: We revised this section title to ‘Statistical Analysis’. We feel as if this section specifically refers to analysis only with details outlined in the order of our primary analysis (with the primary outcome of interest), followed by secondary outcomes of interest) and finally, the post hoc exploratory analyses.

12. Lines 238 to 241 I would delete this paragraph. It adds little to the explanation of your analysis. 

Response: We deleted this paragraph. 

13. Line 363 Thoughts on why the decline in V02 max in phase 3 – wasn’t this somewhat expected?

Response: We agree that declines in Phase 3 were somewhat expected after the intervention. We added details to our interpretation in the discussion where we posit that weekly supervision from the fitness coaches during the Phase 2 intervention may be important for participants to remain engaged in the intervention (and the study) which may account for the decline in benefits seen in Phase 2 (Line 674-676).

14. Table 3. It would be so much clearer if you would add 2 columns that listed the actual values for HR and BP before and after. You are making the reader work too hard to understand your results.

Response: Given this was not a simple pre-post analysis, adding pre and post values to the table would be misleading. In addition, sample sizes differ for each, which would add further confusion. The aim of Tables 2 and 3 are to articulate the slope in each phase hence our preference is to retain the focus of the tables on the change in trend of the outcomes. 

15. Line 408 to 411 Could you please state whether the strength and flexibility improvements are significant.

Response: We added ‘significant’ to the results for strength and flexibility (Line 456).

16. Line 470 You begin to refer to S5 and S7 here but I can’t find where you previously defined these terms. I see later that it refers to the Supplements. Please provide a little explanation here.

Response: We define the acronym ‘Supporting Information’ (S) upon reference to the first supplement and then refer to S throughout the remainder of the manuscript as indicated in the journal formatting guidelines (Line 292).

17. Limitations: Wasn’t your internal validity affected by the wide variation of the types of exercise participants engaged in? IE group vs individual, spinning vs yoga. Did you control statistically for this?

Response: We did not control for different variations of community-based exercise in the study. This was not possible or desirable for the intent of our study. Our intent was to implement a community-based exercise program that included a combination of aerobic, resistive, flexibility and balance exercise. We adopted a ‘guided adaptation’ approach whereby the intervention was adapted based on the ongoing assessment of the needs and ability of participants by the fitness coach (Line 629-632). 

Reviewer #2: 

This is an interesting although dated (enrolment ended in January 2017) real-world evaluation of a community-based exercise (CBE) intervention cardiorespiratory fitness, cardiovascular health, strength, flexibility and physical activity among adults living with HIV seen for care in Toronto Canada.

I have a few remarks mainly about the statistical analysis and interpretation of the results.

Main points

1. The main conclusion seems to be that CBE had no effect on the primary outcome of change in peak VO2. I respectfully disagree with the authors on this point. The estimated change in peak V̇O2 attributed to the six-month phase 2 CBE intervention was 0.56 ml/kg/min (95% Confidence Interval (CI): -1.27, 2.39). Thus although p-value was >0.05 the data are compatible with a 2.39 ml/kg/min increase in peak VO2, which according to the authors is clinically significant (>2.0 ml/kg/min). Therefore, the study is inconclusive as both possible harm and benefit of CBE cannot be ruled out because of low statistical power. Suggest to rephrase text and conclusions accordingly. 

Response: We agree with the reviewer that while the point estimate was not significant, the upper limit of the 95% confidence interval suggests a possible improvement of 2.39 ml/kg/min. We agree results pertaining to VO2peak are inconclusive suggesting both a possible decline and benefit of CBE. We rephrased the abstract, manuscript text, and conclusions accordingly. 

2. The distinction between statistical and clinical significance applies also for the secondary endpoints. Thus, for example, what are the possible clinical implications of a reduction of 5.18 mmHg in systolic blood pressure? How does this translate into a reduction of risk of cardiovascular morbidity and mortality? Authors should consider to estimate these risks by means of the Framingham or D.A.D CVD risk scores. Statistical significance without clinical significance has little relevance. 

Response: We do not have the ability to compute Framingham or DAD CVD risk scores. However, we agree with the importance of further discussing the clinical importance of the BP results and added further discussion about the interpretation of our results in relation to systematic review evidence on the reduction in risk of cardiovascular disease and adverse events. We indicate that the decreasing trend in systolic BP in the study may suggest a trend towards a potential clinically important benefit to cardiovascular health for people living with HIV (Line 646-652). 

3. One key factor that might explain the discrepancies with the results of other studies is the particular case-mix of the target population. Although participants had to be ‘medically stable and safe to participate in exercise activities’ Table 1 indicates that >35% had muscle pain or arthritis at baseline, 19% reported bad health and 82% had at least two other co-morbidities. This issue seems to be confirmed by the average level of peak VO2 at baseline. Is it conceivable that if peak VO2 drops below a certain level CBE is bound to be less effective? Authors should try a sensitivity analysis after excluding the n=20 patients who self-reported bad health at enrolment? 

Response: Given the variability in Vo2peak across and within participants illustrated in Figure 2, there does not appear to be an association between lower baseline scores resulting in lower intervention or follow-up phase Vo2peak scores, indicating less effectiveness of CBE. To further illustrate the variability in VO2peak within participants, we added a Supplemental File with individual trajectory plots of Vo2peak for each participant throughout all phases of the study (See S3 - Supplemental File 3). 

4. Mixed linear model assumes that the outcome has a symmetric (Normal) distribution. Unclear whether this was the case for all endpoints, or whether scale transformations might be needed. Some of the secondary endpoints are discrete scores obtained from a questionnaire (e.g. RAPA, ranging from 1 to 5). Maybe a Poisson model would have been more realistic distribution assumption for these? 

Response: Model assumptions were tested using standardised residual and normal q-q plots for all outcomes (including RAPA aerobic). The RAPA aerobic scores are not counts, and do not have a higher proportion of 0/1 scores and so for this outcome we felt the linear model was more appropriate for our analysis. 

5. I do not follow the hybrid parametrization which was used in the mixed linear model regarding the change of slope over time. The two most commonly used parametrisations are i) the beta coefficients are the estimates of the slope in certain time window or ii) the betas are the estimates of the change in slope in those same windows. The authors appeared to have used an hybrid of these with the betas for the phase 1 and phase 2 following parametrisation i) while the betas for phase 3 following parametrisation ii). I think this is confusing for the reader. I believe the important contrasts are the slope of phase 2 vs. phase 1 and the slope of phase 3 vs. phase 2. Corresponding correct p-values should be given also in Figure 2.

Response: The reviewer is correct, this model parametrization was chosen so that we could incorporate varying intervention durations for participants. In the tables we report the estimated intervention effect as the difference in the phase 2 and phase 1 slope over the six-month intervention. We then report the separate slopes during phases 1 and 2 and the difference in slope for phase 3 vs phase 2. We had incorrectly labelled the Follow-up slope as the difference in slope between “the intervention and baseline phase” this should have read “the difference of the follow-up phase and the intervention phase” and has been corrected in the tables and supplemental files throughout. We revised Figure 2 with the corrected p values. 

6. There is substantial attrition with only 84% completing phase 2 and 77% phase 3. Missing data can introduce bias in mixed linear model analysis. The mentioned paper by Peter et al is a specific scenario in which only peak VO2 is simulated to be missing. If a person drops out from the study, all other relevant covariates will be also missing, starting from the exposure. In this alternative scenario where the imputation model would account for relevant predictor variables that are not incorporated in the mixed model multiple imputation is instead beneficial (see paragraph 4.4 Generalizability in Peter et al). So something better needs to be done to handle missing data. 

Response: We respectfully disagree with the reviewer. We have no time-varying covariates and beyond age and sex, no other predictors that would be appropriate in an imputation model. After much consideration, we felt that imputation of the outcome was inappropriate and that mixed effects modelling, which incorporates all observed data for every participant was the best strategy for modelling the observed data. Any imputation would effectively be making an assumption about the outcome trajectories that we would be unable to support.

7. I would remove the ‘an interrupted time series (ITS)’ description in the title and elsewhere (e.g. abstract line 120, 608). Again I think that could be misleading as ARIMA models are typically used for time series data. However, a different methodology has been implemented here. 

Response: As suggested, we removed reference to ‘interrupted time series’ from the title and manuscript. Instead we referred to intervention study in the title (to align with the journal’s requirements for including the study design in the title). We revise to single group prospective intervention study throughout the remainder of the manuscript. 

8. The study has no control group. I assume that peak VO2 could have changed over time (from phase 1 to phase 2) as effect of simple regression to the mean. How was this effect accounted for in the analysis? 

Response: The reviewer is correct. There is no control group for the study, which is why a three-phase study was implemented. Participants were followed bimonthly over phase 1 (baseline) so that the change in VO2 peak during baseline could be modelled. To estimate an intervention effect we estimated the change due to the intervention by removing the observed baseline trend so that only the difference in trend from phase 2 to phase 1 was considered to be due to the intervention. Our three-phased study enabled us to examine the long-term durability effect of exercise over time when translated into, and influenced by, a real-world setting. 

9. Sampling was conducted on voluntary basis. This could have led to have a selected sample of people with higher self-health awareness, level of education (38% had at least a University degree), etc. Do the authors have characteristics of the whole YMCA population for comparison? Because these factors are also a possible cause of cardiovascular disease, the possibility of having introduced collider bias should be better discussed. 

Response: Voluntary sampling may indeed reduce the generalisability of the findings, but does not introduce collider bias into the estimation of the effect of the CBE on cardiopulmonary fitness, which is based on measurements taken before and after the implementation of the intervention. In Canada, 62% of adults held a university degree in 2018, compared to 44% on average across Organization for Economic Cooperation and Development (OECD) countries, hence 38% of participants with a university education in this study sample was lower than the Canadian average hence we do not expect level of education contributed to the proposed selection bias (OECD, Education at a Glance 2019 - https://www.oecd.org/education/education-at-a-glance/EAG2019_CN_CAN.pdf).

Other points

1. Line 616. RE-AIM framework is introduced here for the first time. Should be spelled out and better described.

Response: We spelled out the acronym in the methods the first time used, which will help with description. For further description of the RE-AIM framework we refer the reader to our protocol. https://bmjopen.bmj.com/content/6/10/e013618#ref-47 (Line 150-151).

2. Lines 272-274. Unclear sentence. If removing sex resulted in a reduced model fit than sex should be retained in the model, not removed? Typo ‘adust’ instead of adjust. 

Response: This was incorrectly phrased and has been corrected. Thank you for indicating the typo, this has been corrected.

3. Line 301. Although it is explained in detail in Supplementary material, what was the minimum improvement in peak VO2 that the authors did not want to miss when calculating the sample size (2.39 ml/kg/min?). This needs to be clearly stated here instead of the generic ‘a trend change’.

Response: We did not use a minimum Vo2peak to calculate sample size. Sample size was based on our original interrupted series time series approach referring to the number of observational time points (outlined in the protocol). For a model of 12 time points to detect level and trend change, assuming an effect size of 1, equal preintervention and postintervention time periods, statistical significance p<0.05 and an autoregression error time series model with lag 1, and autocorrelation estimate of 0.3, we would expect a power of 0.80 (Zhang, 2011; McLeod, 2008). We aimed to recruit ∼120 adults living with HIV, with the goal for 75 participants to complete the study, the number of recommended observations in interrupted time series to achieve acceptable variability of the estimates at each time point (McLeod 2008). This was due to the observed ∼60% retention rate observed an earlier pilot study and was determined as feasible for the YMCA to accommodate administering the fitness assessments and providing weekly coaching during the intervention phase. We added further detail to the manuscript and reference and refer to our published protocol in the manuscript for more details. (Line 330-337)

4. Lines 316-318. Why were minutes of Self-Reported Physical Activity not collected for phase 1? Seem important baseline data given the study design.

Response: We administered the weekly self-reported exercise questionnaire in Phase 2 and 3 only. While we agree that having weekly minutes of physical activity during the 8 month monitoring phase would have enabled comparisons with Phase 1, our aim with the baseline monitoring phase was to establish a ‘business as usual’ context for this phase of the study. Asking participants to self-report their weekly physical activity during the monitoring phase may have influenced participants to increase their physical activity during this phase as well as increase participant burden of measurement in the study. 

5. Page 19. Table 1. I would report in the Table itself that the slopes estimate refer to changes per month. Regarding the legend. A Follow-up slope is the difference in slope between the intervention and baseline phase; this part seems to be incorrect and it is not matching the text in line 364; the slope in phase 3 should be calculated as the sum of the slope in phase 2 (+0.21) and the change from phase 2 vs. phase 3 (-0.37) =-0.16 ml/kg/min/month. I would report this in the table and not -0.37 to improve clarity ( see main point #5 above). 

Response: Thank you for the suggestion. We added ‘change per month’ for the slope estimates directly in Table 2 and the tables in the supplemental files accordingly. Regarding Table 2, we agree the Phase 3: Follow-Up slope was confusing as labelled. We revised this parameter to the following: “Phase 3: Difference in Follow-Up (Phase 3) and Intervention (Phase 2) slopea (change per month).” We also revised the Legend to now correctly state a Estimate refers to the difference in slope between the follow-up and intervention phase. We added further description to the Legend of Table 2: “Phase 3 (Follow-Up) slope is the sum of Phase 2 (intervention) slope and Phase 3 (difference in Follow-Up and Intervention slope): (0.211) + (-0.368) = -0.16 ml/kg/min.” We retained the original estimates in the Tables as this aligns with our interest in reporting whether there was any change in the difference in VO2peak in the follow-up phase related to the intervention phase. We trust the revisions will help to clarify interpretation. We revised the remaining Tables in the Supplemental Files accordingly. 

 6. Line 484. I assume this is referring to the interaction term (duration of treatment phase multiplied by dose). Please rephrase, avoiding statistical jargon 

Response: We revised to “intervention by dose”. (Line 541)

7. Line 526-529. I do not follow, if anything the possible effect of the regression to the mean needs to be accounted for? 

Response: Regression to the mean is the phenomenon of observing a more central observation after observing an extreme observation (ie a test score of 80% after a test score of 100%). Our study repeatedly measured participants over three study phases in order to model the change in VO2 over the study. Any participants with unusually high or low true values would be unlikely to impact the results because it is the change over time that was of interest. Errant high/low observations on a single occasion would also be unlikely to influence the results. 

8. Lines 626-627. The authors make a big deal of the differences in secondary outcomes based on weak evidence (p-value non controlled for multiple comparisons) in Conclusions, Abstract, etc. So this sentence seems contradictory.

Response: We revised the conclusion statement to the following: “Community-dwelling adults living with HIV who engaged in a six-month CBE intervention supervised weekly by a fitness coach demonstrated inconclusive results with respect V̇O2peak, and potential improvements in some outcomes of cardiovascular health, strength, flexibility and self-reported physical activity. (Line 736-738)

9. Lines 632-634. There could be a more general issue of selection bias (see main points #3 and #9 above) and possible implications for internal validity are unclear.

Response: Thank you we added these points to the limitations. “Participants who enrolled in this 22-month study may have been more likely to possess a higher level of self-health awareness compared to the larger HIV population. Generalizability of findings to the broader HIV population, other fitness centres, and in other contexts such as low-and middle income countries is unclear.” (Line 729-732).

Reviewer #3: 

The present study shown in this paper had done a perfect job on reporting a long-term tracking clinical trials by applying exercise intervention on HIV patients. The methods utilized here was appropriate, the results were correct, and the conclusion they finally gain was reliable. At last but at least, they reported that the effect of exercise intervention on HIV therapy was effective, which enlighted patients with hopeful treatment strategy. And again, as they mentioned in the ending, featured exercise therapeutic regimen needed to be elucidated.

Based on its topic and the perfect work they have done, I strongly recommend this paper to be published in PLOS ONE.

Response: Thank you.

---

## [Decision Letter · Decision Letter 1]

26 Aug 2021

PONE-D-21-15069R1

Examining the impact of a community-based exercise intervention on cardiorespiratory fitness, cardiovascular health, strength, flexibility and physical activity among adults living with HIV: a three-phased intervention study

PLOS ONE

Dear Dr. O'Brien,

Thank you for submitting your manuscript to PLOS ONE. After careful consideration, we feel that it has merit but does not fully meet PLOS ONE’s publication criteria as it currently stands. Therefore, we invite you to submit a revised version of the manuscript that addresses the points raised during the review process.

We look forward to receiving your revised manuscript.

Kind regards,

Walid Kamal Abdelbasset, Ph.D.

Academic Editor

PLOS ONE

Journal Requirements:

Reviewers' comments:

Reviewer's Responses to Questions

**Comments to the Author**

1. If the authors have adequately addressed your comments raised in a previous round of review and you feel that this manuscript is now acceptable for publication, you may indicate that here to bypass the “Comments to the Author” section, enter your conflict of interest statement in the “Confidential to Editor” section, and submit your "Accept" recommendation.

Reviewer #1: (No Response)

Reviewer #3: All comments have been addressed

2. Is the manuscript technically sound, and do the data support the conclusions?

Reviewer #1: Yes

Reviewer #3: Yes

3. Has the statistical analysis been performed appropriately and rigorously? 

Reviewer #1: Yes

Reviewer #3: Yes

4. Have the authors made all data underlying the findings in their manuscript fully available?

Reviewer #1: Yes

Reviewer #3: Yes

5. Is the manuscript presented in an intelligible fashion and written in standard English?

Reviewer #1: Yes

Reviewer #3: Yes

6. Review Comments to the Author

Reviewer #1: Please see my attached comments. I have feedback for both the PLOS One editorial team and minor changes for the manuscript authors.

Reviewer #3: The authors completed all the work for this paper, and it is worth publishing. The methods utilized here was appropriate, the results were correct, and the conclusion they finally gain was reliable. At last but at least, they reported that the effect of exercise intervention on HIV therapy was effective, which enlighted patients with hopeful treatment strategy. And again, as they mentioned in the ending, featured exercise therapeutic regimen needed to be elucidated. Based on its topic and the perfect work they have done, I strongly recommend this paper to be published in PLOS ONE.

7. PLOS authors have the option to publish the peer review history of their article (what does this mean?). If published, this will include your full peer review and any attached files.

Reviewer #1: No

Reviewer #3: No

---

## [Author Response · Author response to Decision Letter 1]

28 Aug 2021

Journal Requirements:

Response: Thank you. We reviewed all references and revised the reference list. There were two citations that reflected older citations (e.g. in press) with our referencing software – these have now been revised to reflect the current and accurate citations. In addition, we added reference #39 and #48 to reflect the requested additions from Reviewer 1 below. 

We uploaded new figures after going through the PACE software. We revised the TREND checklist accordance to the new page numbers of the new revised (clean) manuscript. 

Reviewer 1 Comments

Please see my attached comments. I have feedback for both the PLOS One editorial team and minor changes for the manuscript authors.

1. I think the revised manuscript is much stronger and would ask you only to incorporate some of your explanations to me below, into your actual revised manuscript. When I asked for clarifications, it was not so I could personally understand your meaning better, but so you could enlighten your readership on the point in question by adding this information to your revision. So, if you could insert the following passages highlighted in yellow into an appropriate location (typically Methods or Limitations) in the manuscript, I think it would be stronger and more readable.

Previous Review Comment 4. I could not find whether participants were compensated for this? Did I miss that? If not, I wonder if it would have helped participation and statistical significance? It is a big study and I am surprised that compensation was not built in. 

Initial Response: Participants were not compensated for their engagement in the CBE intervention as this would have introduced a co-intervention with incentive to exercise. Participants received weekly personal training / supervision during Phase 2 (intervention) as well as a YMCA membership providing access to the YMCA throughout Phase 2 and Phase 3. Details of receiving a YMCA membership and access to the fitness coach is included in the description of the Intervention. (Line 172-174; Line 197-198). 

Response: Details of what was involved in the intervention (provision of YMCA membership and weekly personal training) are included in the methods of the manuscript (Page 8 and 9). We added details of our rationale pertaining to compensation in the Discussion section (Page 30).

Previous Comment 7. Lines 127-128. I would like to see a short explanation of why you chose your secondary measures, You could have gone in a lot of different directions on this. 

Initial Response: Our study included primary and secondary outcomes of interest measured by objective and self-reported questionnaire assessments. This study reports on the primary outcome of interest (Vo2peak) and self-reported and objective physical health assessments related to the primary outcome of interest. The secondary outcomes focused on objective measures of cardiopulmonary fitness, strength, and flexibility, as they are related to the primary outcome of interest, Vo2peak, and commonly reported in previous systematic reviews on HIV and exercise. (Line 207-209).

Response: The description of primary and secondary outcomes is indicated in the Methods of the manuscript. For instance, the following is included at the beginning of the Materials and Methods section, “Our primary outcome was cardiorespiratory fitness (peak oxygen consumption measured by V̇O2peak). Maximum or peak oxygen consumption (V̇O2peak) is a direct measure of cardiorespiratory fitness.” and “Secondary outcomes included resting heart rate (beats/min), blood pressure (systolic and diastolic; mmHg), strength (upper and lower extremity muscle strength and endurance), flexibility (hamstrings) and self-reported physical activity. These secondary outcomes were chosen because they are related to the primary outcome of interest, V̇O2peak, and commonly reported in previous systematic reviews on HIV and exercise.” The following is also included in the Data Collection section: “We examined the primary outcome of interest (V̇O2peak) and secondary outcomes related to the primary outcome of interest as measured by objective and self-reported physical health assessments.” (Page 6 and 9).

Previous Comment 8. Lines 129. An explanation of why you chose the time periods of 8 mos, 6 mos and 8 mos. Would be helpful 

Initial Response: We chose a 6-month CBE duration because it aligns with transtheoretical model evidence on the stages of behaviour change whereby the ‘action’ of practicing a new behaviour (exercise) lasts up to 24 weeks followed by the ‘maintenance’ stage during which commitment to sustaining the new behavior (self-monitored exercise) is solidified (Prochaska, 1997). The 8 month baseline and 8 month follow ensured equal timeframes prior to and after the intervention, ensured an equal number of data points in each phase (Baseline: T1-T4; Intervention: T5-T8; Follow-Up: T9-12). We include these details in our protocol referenced in the manuscript. 

Response: We add the following to the Intervention Section: “We chose a 6-month CBE duration because it aligns with transtheoretical model evidence on the stages of behaviour change whereby the ‘action’ of practicing a new behaviour (exercise) lasts up to 24 weeks followed by the ‘maintenance’ stage during which to solidify a commitment to sustaining the new behavior (self-monitored exercise).” (Page 8). We added the following to the Materials and Methods Section: “The 8 month baseline and 8 month follow-up ensured equal timeframes prior to and after the intervention, ensured an equal number of data points in each phase (Baseline: T1-T4; Intervention: T5-T8; Follow-Up: T9-12).” (Page 7)

Previous Comment 10. -Lines 226 to 235 Please provide a rationale for the design of this Exercise Dose questionnaire. Was it adapted from an established validated instrument? Similarly the demographics questionnaire 

Initial Response: Our questions about dosage in the demographic questionnaire were guided by the Canadian Physical Activity Guidelines (CPAG), which uses number of minutes engaged in moderate to vigorous aerobic physical activity in the past week as a unit of measurement. This was not from an existing validated instrument. The demographic questionnaire also was not a previous validated instrument. As a team, we identified constructs important for personal, health-related and exercise-related characteristics. We piloted the questionnaire prior to implementation with members of the team. 

Response: We added the above details to the Exercise Dose description: “We developed the weekly exercise, and demographic and health questionnaires as a team to include items that comprised personal, health and exercise-related characteristics. We piloted the weekly exercise questionnaire and demographic and health questionnaire prior to implementation with members of the team.” (Page 12).

Previous Comment 17. Limitations: Wasn’t your internal validity affected by the wide variation of the types of exercise participants engaged in? IE group vs individual, spinning vs yoga. Did you control statistically for this? 

Initial Response: We did not control for different variations of community-based exercise in the study. This was not possible or desirable for the intent of our study. Our intent was to implement a community-based exercise program that included a combination of aerobic, resistive, flexibility and balance exercise. We adopted a ‘guided adaptation’ approach whereby the intervention was adapted based on the ongoing assessment of the needs and ability of participants by the fitness coach (Line 629-632)

Response: We do not consider our lack of control group as a limitation of our study, as this was not our intent given the implementation science study design. We address this in the Discussion –“Our study did not include a comparison group; instead we considered the Phase 1 baseline as a control or monitoring phase of the study from which to estimate an effect.” (Page 29) and later on when we state: “In our study, we adopted a ‘guided adaptation’ approach whereby the intervention was tailored to the ongoing assessment of the needs and ability of participants by the fitness coach…. This aligned with our implementation science approach, to evaluate the impact of the intervention in the ‘real world’ setting.” (Page 30)

Reviewer #3 Comments

The authors completed all the work for this paper, and it is worth publishing. The methods utilized here was appropriate, the results were correct, and the conclusion they finally gain was reliable. At last but at least, they reported that the effect of exercise intervention on HIV therapy was effective, which enlighted patients with hopeful treatment strategy. And again, as they mentioned in the ending, featured exercise therapeutic regimen needed to be elucidated. Based on its topic and the perfect work they have done, I strongly recommend this paper to be published in PLOS ONE.

Response: Thank you

---

## [Decision Letter · Decision Letter 2]

6 Sep 2021

Examining the impact of a community-based exercise intervention on cardiorespiratory fitness, cardiovascular health, strength, flexibility and physical activity among adults living with HIV: a three-phased intervention study

PONE-D-21-15069R2

Dear Dr. O'Brien,

We’re pleased to inform you that your manuscript has been judged scientifically suitable for publication and will be formally accepted for publication once it meets all outstanding technical requirements.

Kind regards,

Walid Kamal Abdelbasset, Ph.D.

Academic Editor

PLOS ONE

Additional Editor Comments (optional):

Reviewers' comments:

Reviewer's Responses to Questions

**Comments to the Author**

1. If the authors have adequately addressed your comments raised in a previous round of review and you feel that this manuscript is now acceptable for publication, you may indicate that here to bypass the “Comments to the Author” section, enter your conflict of interest statement in the “Confidential to Editor” section, and submit your "Accept" recommendation.

Reviewer #1: All comments have been addressed

2. Is the manuscript technically sound, and do the data support the conclusions?

Reviewer #1: Yes

3. Has the statistical analysis been performed appropriately and rigorously? 

Reviewer #1: Yes

4. Have the authors made all data underlying the findings in their manuscript fully available?

Reviewer #1: Yes

5. Is the manuscript presented in an intelligible fashion and written in standard English?

Reviewer #1: Yes

6. Review Comments to the Author

Reviewer #1: All of my requested changes have beeen made. I recommend it be accepted for publication in its current form.

7. PLOS authors have the option to publish the peer review history of their article (what does this mean?). If published, this will include your full peer review and any attached files.

Reviewer #1: No

---

## [Editor Report · Acceptance letter]

15 Sep 2021

PONE-D-21-15069R2 

Examining the impact of a community-based exercise  intervention on cardiorespiratory fitness, cardiovascular health, strength, flexibility and physical activity among adults living with HIV: a three-phased intervention study 

Dear Dr. O'Brien:

I'm pleased to inform you that your manuscript has been deemed suitable for publication in PLOS ONE. Congratulations! Your manuscript is now with our production department. 

Kind regards, 

on behalf of

Dr. Walid Kamal Abdelbasset 

Academic Editor

PLOS ONE